# Beyond BatchNorm: Towards a Unified Understanding of Normalization in Deep Learning

**Ekdeep Singh Lubana**[1*], **Robert P. Dick**[1], **Hidenori Tanaka**[2,3]
[1]EECS Department, University of Michigan
[2]Department of Applied Physics, Stanford University
[3]Physics & Informatics Laboratories, NTT Research, Inc.

## Abstract

Inspired by BatchNorm, there has been an explosion of normalization layers in deep learning. Recent works have identified a multitude of beneficial properties in BatchNorm to explain its success. However, given the pursuit of alternative normalization layers, these properties need to be generalized so that any given layer's success/failure can be accurately predicted. In this work, we take a first step towards this goal by extending known properties of BatchNorm in randomly initialized deep neural networks (DNNs) to several recently proposed normalization layers. Our primary findings follow: (i) similar to BatchNorm, activations-based normalization layers can prevent exponential growth of activations in ResNets, but parametric techniques require explicit remedies; (ii) use of GroupNorm can ensure an informative forward propagation, with different samples being assigned dissimilar activations, but increasing group size results in increasingly indistinguishable activations for different samples, explaining slow convergence speed in models with LayerNorm; and (iii) small group sizes result in large gradient norm in earlier layers, hence explaining training instability issues in Instance Normalization and illustrating a speed-stability tradeoff in GroupNorm. Overall, our analysis reveals a unified set of mechanisms that underpin the success of normalization methods in deep learning, providing us with a compass to systematically explore the vast design space of DNN normalization layers.

## 1 Introduction

Normalization techniques are often necessary to effectively train deep neural networks (DNNs) [1, 2, 3]. Arguably, the most popular of these is BatchNorm [1], whose success can be attributed to several beneficial properties that allow it to stabilize a DNN's training dynamics: for example, ability to propagate informative activation patterns in deeper layers [4, 5]; reduced dependence on initialization [6, 7, 8]; faster convergence via removal of outlier eigenvalues [9, 10]; auto-tuning of learning rates [11], equivalent to modern adaptive optimizers [12]; and smoothing of loss landscape [13, 14]. However, depending on the application scenario, BatchNorm's use can be of limited benefit or even a hindrance: for example, BatchNorm struggles when training with small batch-sizes [3, 15]; in settings with train-test distribution shifts, BatchNorm can undermine a model's accuracy [16, 17]; in meta-learning, it can lead to transductive inference [18]; and in adversarial training, it can hamper accuracy on both clean and adversarial examples by estimating incorrect statistics [19, 20].

To either address specific shortcomings or to replace BatchNorm in general, several recent works propose alternative normalization layers (interchangeably called normalizers in this paper). For example, Brock et al. [23] propose to match BatchNorm's forward propagation behavior in Residual

---

Email: {eslubana, dickrp}@umich.edu, and hidenori.tanaka@ntt-research.com
*Work partially performed during an internship at Physics & Informatics Laboratories, NTT Research.

35th Conference on Neural Information Processing Systems (NeurIPS 2021).

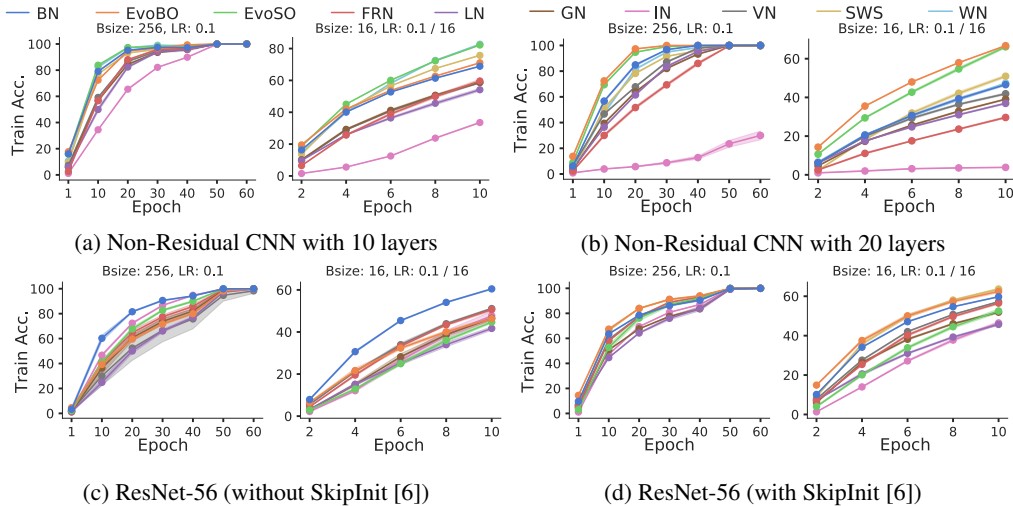

(a) Non-Residual CNN with 10 layers

(b) Non-Residual CNN with 20 layers

(c) ResNet-56 (without SkipInit [6])

(d) ResNet-56 (with SkipInit [6])

Figure 1: **Each normalization method has its own success and failure modes.** We plot training curves (3 seeds) for different combinations of *normalizer* (see Table 1), *network architecture,* and *batch-size* at largest stable initial learning rate on CIFAR-100. Learning rate is scaled linearly with batch-size [21]. Layers for which loss reaches infinity are not plotted. Test curves and several other settings are provided in the appendix. The plots show that all methods, including BatchNorm (BN), have their respective success and failure modes: e.g., LayerNorm (LN) [2] often converges slowly and Instance Normalization (IN) [22] can have unstable training with large depth or small batch-sizes.

networks [24] by replacing it with Scaled Weight Standardization [25, 26]. Wu and He [3] design GroupNorm, a batch-independent method that groups multiple channels in a layer to perform normalization. Liu et al. [27] use an evolutionary algorithm to search for both normalizers and activation layers. Given the right training configuration, these works show their proposed normalizers often achieve similar test accuracy to BatchNorm and even outperform it on some benchmarks. This begs the question, are we ready to replace BatchNorm? To probe this question, we plot training curves for models defined using different combinations of *normalizer, network architecture, batch size,* and *learning rate* on CIFAR-100. As shown in Figure 1, clear trends begin to emerge. For example, we see LayerNorm [2] often converges at a relatively slower speed; Weight Normalization [28] cannot be trained at all for ResNets (with and without SkipInit [6]); Instance Normalization [22] results in unstable training in deeper non-residual networks, especially with small batch-sizes. Overall, evaluating hundreds of models in different settings, we see evident success/failure modes exist for all normalization techniques, including BatchNorm.

As we noted before, prior works have established several properties to help explain such success/failure modes for the specific case of BatchNorm. However, given the pursuit of alternative normalizers in recent works, these properties need to be generalized so that one can accurately determine how normalization techniques beyond BatchNorm affect DNN training. In this work, we take a first step towards this goal by extending known properties of BatchNorm *at initialization* to several alternative normalization techniques. As we show, these properties are highly predictive of a normalizer's influence on DNN training and can help ascertain exactly when an alternative technique is capable of serving as a replacement for BatchNorm. Our contributions follow.

- **Stable Forward Propagation:** In Section 3, we show activations-based normalizers are provably able to prevent exploding variance of activations in ResNets, similar to BatchNorm [5, 6]. Parametric normalizers like Weight Normalization [28] do not share this property; however, we explain why architectural modifications proposed in recent works [6, 7] can resolve this limitation.

- **Informative Forward Propagation:** In Section 4, we first show the ability of a normalizer to generate dissimilar activations for different inputs is a strong predictor of optimization speed. We then extend a known result for BatchNorm to demonstrate the rank of representations in the deepest layer of a Group-normalized [3] model is at least $\Omega(\sqrt{\text{width}/\text{Group Size}})$. This helps us illustrate how use of GroupNorm can prevent high similarity of activations for different inputs if the group size is small, i.e., the number of groups is large. This suggests Instance Normalization [22] (viz.,

GroupNorm with group size equal to 1) is most likely and LayerNorm [2] (viz., GroupNorm with group size equal to layer width) is least likely to produce informative activations.

- **Stable Backward Propagation:** In Section 5, we show normalization techniques that rely on individual sample and/or channel statistics (e.g., Instance Normalization [22]) suffer from an exacerbated case of gradient explosion [29], often witnessing unstable backward propagation. We show this behavior is mitigated by grouping of channels in GroupNorm, thus demonstrating a speed–stability trade-off characterized by group size.

**Related Work:** Due to its ubiquity, past work has generally focused on understanding BatchNorm [5, 4, 6, 9, 10, 7, 13, 29, 30, 31]. A few works have studied LayerNorm [32, 33], due to its relevance in natural language processing. In contrast, we try to analyze normalization methods in deep learning in a general manner. As we show, we can identify properties in BatchNorm that readily generalize to other normalizers and are often predictive of the normalizer's impact on training. Our analysis is inspired by a rich body of work focused on understanding randomly initialized DNNs [34, 35, 36, 37, 38]. Most related to us is the contemporary work by Labatie et al. [39], who analyze the impact of different normalization layers on expressivity of activations and conclude LayerNorm leads to high similarity of activations in deeper layers. As we discuss, this result is in fact a special case of our Claim 3.

## 2 Preliminaries: Normalization Layers for DNNs

We first clarify the notations and operations used by the normalizers discussed in this work. Specifically, we define operators $\mu_{\{d\}}(\mathcal{T})$ and $\sigma_{\{d\}}(\mathcal{T})$, which calculate the mean and standard deviation of a tensor $\mathcal{T}$ along the dimensions specified by set $\{d\}$. $\|\mathcal{T}\|$ denotes the $\ell_2$ norm of $\mathcal{T}$. $\mathrm{RMS}_{\{d\}}(\mathcal{T})$ denotes the root mean square of $\mathcal{T}$ along dimensions specified by set $\{d\}$. For example, for a vector $\mathbf{v} \in \mathbb{R}^n$, we have $\mathrm{RMS}_{\{1\}}(v) = \sqrt{\sum_i v_i^2 / n}$. We assume the outputs of these operators broadcast as per requirements. $\rho(.)$ denotes the sigmoid function. We define symbols $b$, $c$, $x$ to denote the batch, channel, and spatial dimensions. For feature maps in a CNN, $x$ will include both the height and the width dimensions. The notation $c/g$ denotes division of $c$ neurons (or channels) into groups of size $g$. When grouping is performed, each group is normalized independently.

**Normalization Layers:** We analyze ten normalization layers in this work. These layers were chosen to cover a broad range of ideas: e.g., activations-based layers [1, 40], parametric layers [23, 28], hand-engineered layers [3], AutoML designed layers [27], and layers [22, 2, 4] that form building blocks of recent techniques [41].

| Activations-Based Layers $\mu_{\{d\}} = \mu_{\{d\}}(\mathcal{A}); \sigma_{\{d\}} = \sigma_{\{d\}}(\mathcal{A})$ | |
| --- | --- |
| BN [1] | $\frac{\mathcal{A} - \mu_{\{b,x\}}}{\sigma_{\{b,x\}}}$ |
| LN [2] | $\frac{\mathcal{A} - \mu_{\{c,x\}}}{\sigma_{\{c,x\}}}$ |
| IN [22] | $\frac{\mathcal{A} - \mu_{\{x\}}}{\sigma_{\{x\}}}$ |
| GN [3] | $\frac{\mathcal{A} - \mu_{\{c/g,x\}}}{\sigma_{\{c/g,x\}}}$ |
| FRN [40] | $\frac{\mathcal{A}}{\mathrm{RMS}_{\{x\}}}$ |
| VN [4] | $\frac{\mathcal{A}}{\sigma_{\{b,x\}}}$ |
| EvoBO [27] | $\frac{\mathcal{A}}{\max\{\sigma_{\{b,x\}}, v \odot \mathcal{A} + \sigma_{\{x\}}\}}$ |
| EvoSO [27] | $\frac{\mathcal{A}\rho(v \odot \mathcal{A})}{\sigma_{\{c/g,x\}}}$ |

| Parametric Layers $\mu_{\{d\}} = \mu_{\{d\}}(\mathcal{W}); \sigma_{\{d\}} = \sigma_{\{d\}}(\mathcal{W})$ | |
| --- | --- |
| WN [28] | $g\frac{\mathcal{W}}{\|\mathcal{W}\|}$ |
| SWS [23] | $g\frac{\mathcal{W} - \mu_{\{c,h,w\}}}{\sigma_{\{c,x\}}}$ |

Table 1: **Operations performed by different normalizers.** $\mathcal{A}$ denotes layer input; $\mathcal{W}$ denotes incoming neuron weights to a neuron.

1. *Activations-Based Layers*: BatchNorm (BN) [1], Layer-Norm (LN) [2], Instance Normalization (IN) [22], GroupNorm (GN) [3], Filter Response Normalization (FRN) [40], Variance Normalization (VN) [4], EvoNormBO [27], and EvoNoRMSO [27] fall in this category. These layers function in the activation space. Note that Variance Normalization is an ablation of BatchNorm that does not use the mean-centering operation. Typically, given activations $\mathcal{A}_L$ at layer $L$, these layers use an operation of the form $\mathcal{A}_{\mathrm{norm}} = \phi\left(\frac{\gamma}{\sigma_{\{d\}}(\mathcal{A}_L)}(\mathcal{A}_L - \mu_{\{d\}}(\mathcal{A}_L)) + \beta\right)$. Here, $\gamma$ and $\beta$ are learned affine parameters used for controlling quantities affected by the normalization operations (such as mean, standard deviation, and RMS) and $\phi$ is a non-linearity, such as ReLU. The exact operations for these layers, minus the affine parameters, are shown in Table 1.

2. *Parametric Layers*: Weight Normalization (WN) [28] and Scaled Weight Standardization (SWS) [23] fall in this category. Table 1 shows the exact operations. These layers function in the parameter space and act on a filter's weights ($\mathcal{W}$) to generate normalized weights ($\mathcal{W}_{\mathrm{norm}}$). The normalized weights $\mathcal{W}_{\mathrm{norm}}$ are used for processing the input: $\mathcal{A}_{L+1} = \phi(\mathcal{W}_{\mathrm{norm}} * \mathcal{A}_L)$.

# 3 Stable Forward Propagation

Stable forward propagation is a necessary condition for successful DNN training [36]. In this section, we identify and demystify the role of normalization layers in preventing the problem of *exploding* or *vanishing activations* during forward propagation. These problems can result in training instability due to exploding or vanishing gradients during backward propagation [36, 38]. Building on a previous study on BatchNorm, we first show that activations-based normalizers provably avoid exponential growth of variance in ResNets[1], ensuring training stability. Thereafter, we show parametric normalizers do not share this property and ensuring stable training requires explicit remedies.

## 3.1 Activations-Based Normalizers and Exponential Variance in Residual Networks

Hanin and Rolnick [38] show that for stable forward propagation in ResNets, the average variance of activations should not grow exponentially (i.e., should not explode). Interestingly, Figure 1 shows that all activations-based normalizers are able to train the standard ResNet [24] architecture stably. For BatchNorm, this behavior is provably expected. Specifically, De and Smith [6] find that to ensure variance along the batch-dimension is 1, BatchNorm rescales the $L^{\text{th}}$ layer's residual path output by a factor of $\mathcal{O}\left(1/\sqrt{L}\right)$. This causes the growth of variance in a Batch-Normalized ResNet to be linear in depth, hence avoiding exponential growth of variance in and ensuring effective training. We now show this result can be extended to other normalization techniques too.

**Claim 1.** *Similar to BatchNorm [6], GroupNorm [3] avoids exponential growth of variance in ResNets, ensuring stable training.*

*Proof.* We follow the same setup as De and Smith [6]. Assume the $L^{\text{th}}$ residual path ($f_L$) is processed by a normalization layer $\mathcal{N}$, after which it combines with the skip connection to generate the next output: $\mathbf{y}_L = \mathbf{y}_{L-1} + \mathcal{N}(f_L(\mathbf{y}_{L-1}))$. The covariance of layer input and Residual path's output is assumed to be zero. Hence, the output's variance is: $\text{Var}(\mathbf{y}_L) = \text{Var}(\mathbf{y}_{L-1}) + \text{Var}(\mathcal{N}(f_L(\mathbf{y}_{L-1})))$. Now, assume GroupNorm with group size $G$ is used for normalizing the $D$-dimensional activation signal, i.e., $\mathcal{N} = \text{GN}(.)$. This implies for the $g^{\text{th}}$ group, $\sigma_{g,x}(GN(f_L(\mathbf{y}_{L-1}))) = 1$. Then, for a batch of size $N$, denoting the $i^{\text{th}}$ sample activations as $\mathbf{y}_L^{(i)}$, and using $(\mathbf{y}_L^{(i)})^j$ to index the activations, we note the residual output's variance averaged along the spatial dimension is: $\langle \text{Var}(\mathcal{N}(f_L(\mathbf{y}_{L-1}))) \rangle = \frac{1}{D}\sum_{j=1}^{D}(\frac{1}{N}\sum_{i=1}^{N}(\text{GN}(f_L(\mathbf{y}_{L-1}^{(i)})^j)^2) = \frac{1}{N}\sum_{i=1}^{N}(\frac{1}{D}\sum_{j=1}^{D}(\text{GN}(f_L(\mathbf{y}_{L-1})^{(i)})^j)^2) = \frac{1}{N}\sum_{i=1}^{N}\frac{G}{D}(\sum_{g=1}^{D/G}(\sigma_{g,x}(\text{GN}(f_L(\mathbf{y}_{L-1})^{(i)})))^2) = 1$. Overall, this implies $\langle \text{Var}(\mathbf{y}_L) \rangle = \langle \text{Var}(\mathbf{y}_{L-1}) \rangle + 1$. Recursively applying this relationship for a bounded variance input, we see average variance at the $L^{\text{th}}$ layer is in $\mathcal{O}(L)$. Thus, similar to BatchNorm, use of GroupNorm will enable stable forward propagation in ResNets by ensuring signal variance grows linearly with depth. $\square$

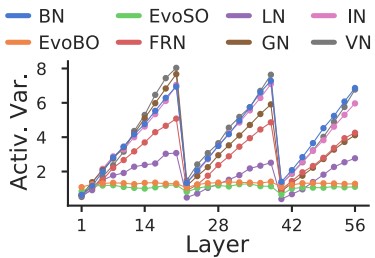

Figure 2: **Activations-based normalizers ensure linear and stable forward propagation, verifying Claim 1.** Activation Variance (Activ. Var.) as a function of layer number in a ResNet-56 [24] processing CIFAR-100 samples.

To understand the relevance of the above result, note that for $G = 1$, GroupNorm is equal to Instance Normalization [22] and for $G = D$, GroupNorm is equal to LayerNorm [2]. Further, since the mean of the signal is assumed to be zero, the average variance along the spatial dimension is equal to the $\text{RMS}_x$ operation used by Filter Response Normalization [40]. Thus, by proving the above result for GroupNorm, we are able to show alternative activations-based normalizers listed in Table 1 also avoid the exponential growth of activation variance in ResNets.

We show empirical demonstrations of Claim 1 in Figure 2, where the average activation variance is plotted for a ResNet-56. As can be seen, for all activations-based normalizers, the growth of variance is linear in the number of layers. At the end of a Residual module, which spatially downsamples the signal, the variance plummets. However, the remaining layers follow a pattern of linear growth, as expected by our result. We note our

---

[1]The case of *non-residual* networks is discussed in appendix. In brief, most normalizers help avoid exploding/vanishing activations by enforcing unit activation variance in the batch, channel, or spatial dimensions.

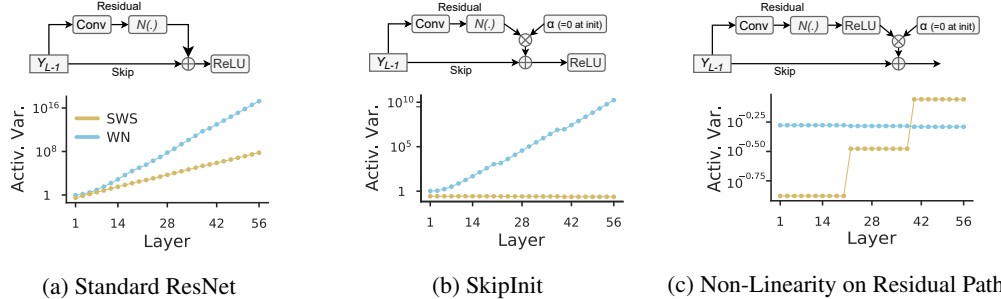

Figure 3: **Parametric normalizers witness exponentially growing variance, verifying Claim 2, but we can stabilize it by modifying the residual-path.** We plot log activation variance as a function of layer number in a randomly initialized ResNet-56 [24], using CIFAR-100 samples, with Scaled Weight Standardization (SWS) [23] and Weight Normalization (WN) [28] for different architectures (simplified illustrations provided on top). (a) *Standard ResNet:* Both SWS and WN witness variance explosion in a standard ResNet model, as claimed in Claim 2. (b) *SkipInit:* SkipInit [6] multiplies the residual signal with a scalar $\alpha$ initialized as zero, thus preventing variance explosion in an SWS model at initialization. Meanwhile, by scaling the non-linearity after addition, a WN model continues to witness exploding variance. (c) *Non-Linearity on Residual Path:* Shifting the non-linearity to the residual path prevents variance explosion in both WN and SWS models.

theory does not apply to EvoNorms, which are designed via AutoML. However, empirically, we see EvoNorms also avoid exponential growth of variance in ResNets. *Thus, our analysis shows, all activations-based normalizers in Table 1 share the beneficial property of stabilizing forward propagation in ResNets, similar to BatchNorm.*

## 3.2 Parametric Normalizers and Exploding Variance in Residual Networks

By default, parametric normalizers such as Weight Normalization [28] and Scaled Weight Standardization [23] do not preserve the variance of a signal during forward propagation, often witnessing vanishing activations. To address this limitation, properly designed output scale and bias corrections are needed. Specifically, for Weight Normalization and ReLU non-linearity, Arpit et al. [42] show the output should be modified as follows: $\mathcal{A}_{L+1} = \sqrt{2\pi/\pi-1}(\phi(\mathcal{W}_{\text{norm}} * \mathcal{A}_L) - \sqrt{1/2\pi})$. For Scaled Weight Standardization, only output scaling is needed [23]: $\mathcal{A}_{L+1} = \phi(\sqrt{2\pi/\pi-1}\mathcal{W}_{\text{norm}} * \mathcal{A}_L)$.

In Figure 1, ResNet training curves for Weight Normalization [28] and Scaled Weight Standardization [23] were not reported as the loss diverges to infinity. As we explain in the following, this is a result of using correction factors designed to enable variance preservation in non-residual networks.

**Claim 2.** *Unlike BatchNorm [6], Weight Normalization [28] and Scaled Weight Standardization [23] witness unstable training due to exponential growth of variance in standard ResNets [24].*

*Proof.* Using the correction factors above, both Weight Normalization and Scaled Weight Standardization will ensure signal variance is preserved on the residual path: $\text{Var}(\mathcal{N}(f(\mathbf{y}_{L-1}))) = \text{Var}(\mathbf{y}_{L-1})$. Thus, using these methods, the output variance at layer $L$ becomes: $\text{Var}(\mathbf{y}_L) = \text{Var}(\mathbf{y}_{L-1}) + \text{Var}(\mathcal{N}(f(\mathbf{y}_{L-1}))) = 2\,\text{Var}(\mathbf{y}_{L-1})$. Recursively applying this relationship for a bounded variance input, we see signal variance at the $L^{\text{th}}$ layer is in $\mathcal{O}(2^L)$. Thus, Weight Normalization and Scaled Weight Standardization witness exponential growth in variance. $\square$

More generally, the above result shows if the residual path is variance preserving, ResNets will witness exploding variance with growing depth. Prior works [43, 5, 8, 6, 7, 44] have noted this result in the context of designing effective ResNet initializations. *Here, we extended this result to show why Weight Normalized and Scaled Weight Standardized ResNets undergo unstable forward propagation.* Empirical demonstration is provided in Figure 3a.

In their work introducing Scaled Weight Standardization [23], Brock et al. are able to circumvent exponential growth in variance by using SkipInit [6]. Specifically, inspired by the fact that BatchNorm biases Residual paths to identity functions, De and Smith [6] propose SkipInit, which multiplies the output of the residual path by a learned scalar $\alpha$ that is initialized to zero. This suppresses the Residual path's contribution, hence avoiding exponential growth in variance (see Figure 3b). Interestingly, even

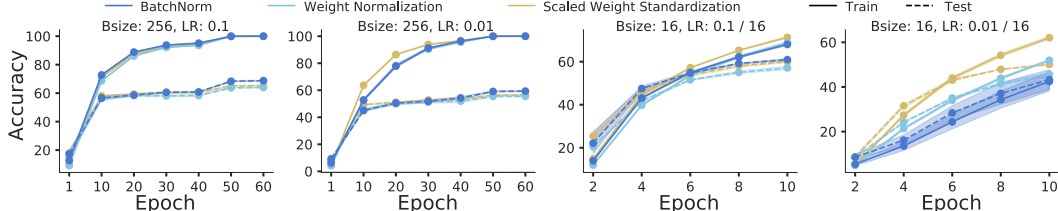

Figure 4: **Modified residual-path allows for successful training with parametric layers.** We plot train/test accuracy (over 3 seeds) for ResNet-56 architecture on CIFAR-100 with non-linearity located on the residual path. We see parametric normalizers can train effectively if scaled non-linearities are not located after the addition operation in a ResNet.

after using SkipInit, we find Weight Normalized ResNets witness variance explosion (see Figure 3b). To explain this behavior, we note that in the standard ResNet architecture, the non-linearity is located after the addition operation of skip connection and the residual path's signals (see Figure 3a). Thus, even if SkipInit is used to suppress the residual path, the non-linearity will still be applied to the skip connections. Since the scale correction for Weight Normalization ($\sqrt{2\pi/\pi-1}$) is greater than 1, this implies the signal output is amplified at every layer to preserve signal variance; however, since convolutions are absent on the skip path, signal variance never decays. Consequently, *variance is only amplified*, causing the variance to increase exponentially in the number of layers (see Figure 3b).

**Training ResNets with Weight Normalization:** The above discussion shows that for Weight Normalization, since the output has to be scaled-up to preserve signal variance, standard ResNets [24] witness exploding activations. This also hints at a solution: place the non-linearity on the Residual path. This modification (see Figure 3c) in fact results in one of the architectures proposed by He et al. in their original work on ResNets [45]. We verify the effectiveness of this modification in Figure 3c. As can be seen, the signal variance in a Weight Normalized ResNet stays essentially constant for this architecture. Furthermore, we show in Figure 4 that these models are able to match BatchNorm in performance for several training configurations. In general, our discussion here explains *the exact reasons why architectures with non-linearity on residual path are better suited for parametric normalizers.* Finally, we note that another ResNet architecture which boasts non-linearity on residual paths is pre-activation ResNets [45]. In their experimental setup for designing Scaled Weight Standardization [23], Brock et al. specifically focused on pre-activation ResNets [45]. This is another reason why the problem of exploding activations does not surface in their work.

## 4 Informative Forward Propagation

Proper magnitude of activations is a necessary, but not sufficient, condition for successful training. Here, we study another failure mode for forward propagation, *rank collapse*, where activations for different input samples become indistinguishably similar in deeper layers. This can significantly slow training as the gradient updates no longer reflect information about the input data [4]. To understand this problem's relevance, we first show why the ability to generate dissimilar activations is useful in the context of normalization methods for deep learning. Specifically, given a randomly initialized network that uses a specific normalizer, we relate its average cosine similarity of activations at the penultimate layer (i.e., layer before the linear classifier) with its mean training accuracy ($= \frac{\sum_{i=1}^{\text{\# of epochs}} \text{Train Acc.}[i]}{\text{\# of epochs}}$, a measure of optimization speed [46]). Results for three different architectures (Non-Residual CNN with 10 layers and 20 layers as well as ResNet-56 without SkipInit) are shown in Figure 5. As can be seen, the correlation between mean training accuracy and the average cosine similarity of activations is high. *In fact, for any given network architecture, one can predict which normalizer will enable the fastest convergence without even training the model.* This shows normalizers which result in more dissimilar representations at initialization are likely to be more useful for training DNNs.

We now note another interesting pattern in Figure 5: *LayerNorm results in highest similarity of activations for any given architecture*. To explain this, we again revisit known properties of Batch-Norm. As shown by Daneshmand et al. [4, 47], BatchNorm provably ensures activations generated by a randomly initialized network have high rank, i.e., different samples have sufficiently different activations. To derive this result, the authors consider activations for $N$ samples at the penultimate layer, $Y \in \mathbb{R}^{\text{width} \times N}$, and define the covariance matrix $YY^T$, whose rank is equal to that of the

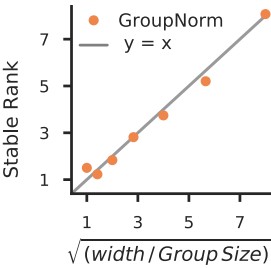

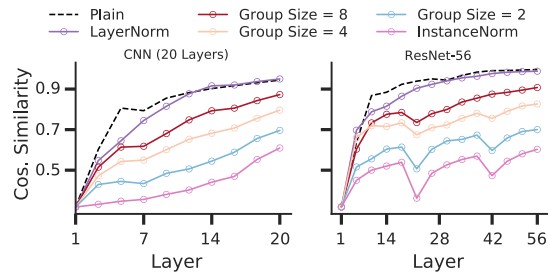

(a) Stable rank vs. group size.

(b) Layer-wise Cosine Similarity.

Figure 6: **The smaller the group size, the higher the rank of the activations, verifying Claim 3** (a) We plot stable rank of activations at the penultimate layer for random Gaussian inputs. As proposed in Claim 3, we find a perfect linear fit between stable rank and values of $\sqrt{\text{Width}/\text{Group Size}}$ for different group sizes. (b) Implications of Claim 3 on CIFAR-100 sampels: by increasing group size (constant across layers), we see similarity of features at any given layer increases. This shows LayerNorm [2] cannot generate informative features, thus witnessing slow convergence (see Figure 5).

similarity matrix $Y^T Y$. The authors then show that in a zero-mean, randomly initialized network with BatchNorm layers, the covariance matrix will have a rank at least as large as $\Omega(\sqrt{\text{width}})$. That is, there are at least $\Omega(\sqrt{\text{width}})$ distinct directions that form the basis of the similarity matrix, hence indicating the model is capable of extracting informative activations. In the following, we propose a claim that extends this result to activations-based normalizers beyond BatchNorm.

**Claim 3.** *For a zero-mean, randomly initialized network with GroupNorm [3] layers, the penultimate layer activations have a rank of at least $\Omega(\sqrt{\text{width}/\text{Group Size}})$, where width denotes the layer-width (e.g., number of channels in a CNN).*

The intuition behind the above claim is based on the proof by Daneshmand et al. [4]. In their work, the authors extend a prior result from random matrix theory which suggests multiplication of several zero-mean, randomly initialized gaussian matrices will result in a rank-one matrix [10]. The use of BatchNorm ensures that on multiplication with a randomly initialized weight matrix, the values of on-diagonal elements of the covariance matrix $YY^T$ are preserved, while the off-diagonal elements are suppressed. This leads to a lower bound of the order of $\Omega(\sqrt{\text{width}})$ on the stable rank [48] of the covariance matrix. Now, if one directly considers the similarity matrix $Y^T Y$ and uses GroupNorm instead of BatchNorm, then a similar preservation and suppression of on- and off-diagonal *matrix blocks* should occur. Here, the block size will be equal to the Group size used for GroupNorm. This indicates the lower bound is in $\Omega(\sqrt{\text{width}/\text{Group Size}})$.

We provide demonstration of this claim in Figure 6a. We use a similar setup as Daneshmand et al. [4], randomly initializing a CNN with constant layer-width (64) and 30 layers. A GroupNorm layer is placed before every ReLU layer and the group size is sweeped from 1 to 64. As seen in Figure 6a, we find a perfect linear fit between the stable rank and the value of $\sqrt{\text{width}/\text{Group Size}}$, validating our claim empirically as well.

To understand the significance of Claim 3, note that the result shows if the group size is large, then use of GroupNorm cannot prevent collapse of representations (i.e., cannot result

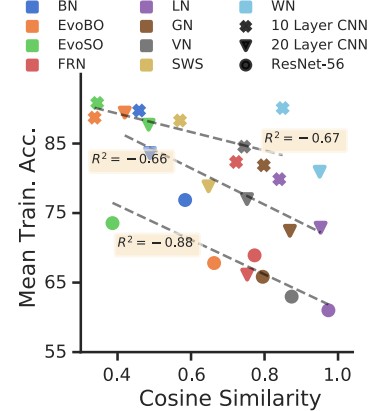

Figure 5: **Informative forward propagation results in faster optimization.** We plot mean training accuracy $(= \frac{\sum_{i=1}^{\text{\# of epochs}} \text{Train Acc.[i]}}{\text{\# of epochs}})$ on CIFAR-100 vs. average cosine similarity at initialization. As shown, normalizers which induce dissimilar activations converge faster. Instance Normalization was removed due to training instability (see Section 5).

in informative activations). To demonstrate this effect, we calculate the mean cosine similarity of activations between different samples of a randomly initialized network that uses GroupNorm. We sweep the group size from layer-width to 1, thus covering the spectrum from LayerNorm (Group Size

= layer-width) to Instance Normalization (Group Size = 1). We analyze both a non-residual CNN with 20 layers and a ResNet-56. Results are shown in Figure 6b and confirm our claim that by grouping the entire layer for normalization, LayerNorm results in highly similar activations. *This explains the slow convergence behavior of LayerNorm in Figure 5.* Meanwhile, if we reduce the group size, similarity of representations decreases as well, indicating generation of informative activations. This shows use of GroupNorm with group size greater than layer-width can help prevent a collapse of features onto a single representation. Importantly, this result helps explain why GroupNorm can serve as a successful replacement for BatchNorm in similarity based self-supervised learning frameworks [49], which often witness representation collapse issues [50]. Similar to BatchNorm, GroupNorm helps discriminate between representations of different inputs, helping avoid a collapse of representations.

## 5   Stable Backward Propagation

Taking the results of Section 4 to the extreme should imply Instance Normalization (i.e., Group Size = 1) is the best configuration for GroupNorm, but as we noted in Figure 1, Instance Normalization witnesses unstable training. To explain this, we describe a "speed-stability" trade-off in GroupNorm in the next section by extending the property of gradient explosion in BatchNorm to alternative normalization layers. Specifically, Yang et al. [29] recently show that gradient norm in earlier layers of a randomly-initialized BatchNorm network increases exponentially with increasing model depth (see Figure 8). This shows the maximum depth of a model trainable with BatchNorm is finite. The theory leading to this result is quite involved, but a much simpler analysis can not only explain this phenomenon accurately, but also illustrate the existence of gradient explosion in alternative layers.

**Gradient explosion in BatchNorm:** Following Luther [51], we analyze the origin of gradient explosion based on the expression of gradient backpropagated through a BatchNorm layer. We calculate the gradient of loss w.r.t. activations at layer $L$, denoted as $\mathbf{Y}_L \in \mathbb{R}^{d_L \times N}$. We define two sets of intermediate variables: (i) pre-activations, generated by weight multiplication, $X_L = W_L Y_{L-1}$ and (ii) normalized pre-activations, generated by BatchNorm, $\hat{X}_L = \mathrm{BN}(X_L) = \frac{\gamma}{\sigma_{\{N\}}(X_L)}(X_L - \mu_{\{N\}}(X_L)) + \beta$. Under these notations, the gradient backpropagated from layer $L$ to layer $L-1$ is (see appendix for derivation): $\nabla_{\mathbf{Y}_{L-1}}(J) = \frac{\gamma}{\sigma_{\{N\}}(X_L)} \mathcal{W}_L^T \mathcal{P}[\nabla_{\hat{\mathbf{X}}_L}(J)]$. Here $\mathcal{P}$ is a composition of two projection operators: $\mathcal{P}[\mathbf{Z}] = \mathcal{P}_{\mathbf{1}_N}^{\perp}[\mathcal{P}_{\mathrm{Ob}(\hat{\mathbf{x}}_L/\sqrt{N})}^{\perp}[\mathbf{Z}]]$. The operator $\mathcal{P}_{\mathrm{Ob}(\hat{\mathbf{x}}_L/\sqrt{N})}^{\perp}[\mathbf{Z}] = \mathbf{Z} - \frac{1}{N}\mathrm{diag}(\mathbf{Z}\hat{\mathbf{X}}_L^T)\hat{\mathbf{X}}_L$ subtracts its input's component that is inline with the BatchNorm outputs via projection onto the Oblique manifold $\mathrm{diag}(\frac{1}{N}\hat{\mathbf{X}}_L\hat{\mathbf{X}}_L^T) = \mathrm{diag}(\mathbf{1})$. Similarly, $\mathcal{P}_{\mathbf{1}}^{\perp}[\mathbf{Z}] = \mathbf{Z}(I - \frac{1}{N}\mathbf{1}_N\mathbf{1}_N^T)$ mean-centers its input along the batch dimension via projection onto $\mathbf{1}_N \in \mathbb{R}^N$.

Notice that at initialization, the gradient is unlikely to have a large component along specific directions such as the all-ones vector ($\mathbf{1}$) or the oblique manifold defined by $\hat{\mathbf{X}}_L$. Thus, the gradient norm will remain essentially unchanged when propagating through the projection operation ($\mathcal{P}$). However, the next operation, multiplication with $\frac{\gamma}{\sigma_{\{N\}}(X_L)}$ ($= \frac{1}{\sigma_{\{N\}}(X_L)}$ at initialization) will re-scale the gradient norm according to the standard deviation of pre-activations along the batch dimension. As shown by Luther [51], for a standard, zero-mean Gaussian initialization, the pre-activations have a standard deviation equal to $\sqrt{\pi-1/\pi} < 1$. Thus, at initialization, the division by standard deviation operation *amplifies* the gradient during backward propagation. For each BatchNorm layer in the model, such an amplification of the gradient will take place, hence resulting in an exponential increase

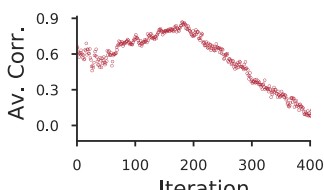

Figure 7: **Gradient norm vs. pre-activation statistics.** We see high correlation between gradient norm and inverse product of layer-wise pre-activation std. deviations.

in the gradient norm at earlier layers. Overall, our analysis exposes an interesting tradeoff in BatchNorm: *Divison by standard deviation during forward propagation, which is important for generating dissimilar activations [4], results in gradient explosion during backward propagation, critically limiting the maximum trainable model depth!* Empirically, the above analysis is quite accurate near initialization. For example, in Figure 7, we show that the correlation between the norm of the gradient at a layer ($\|\nabla_{\mathbf{Y}_L}(J)\|$) and the inverse product of standard deviation of the pre-activations of layers ahead of it ($\Pi_{l=10}^{L+1} 1/\sigma_{\{N\}}(\mathbf{x}_L)$) remains very high (0.6–0.9) over the first few hundred iterations in a 10-layer CNN trained on CIFAR-100.

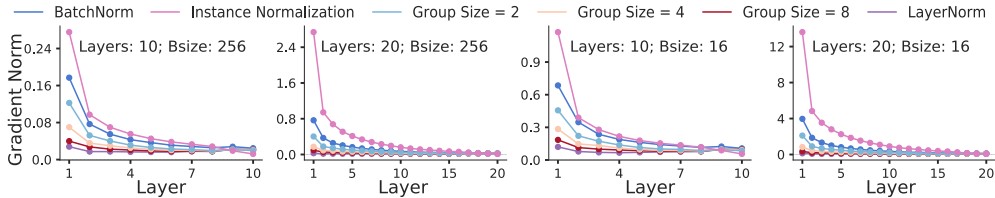

Figure 8: **Small group size increases gradient explosion, verifying Claim 4.** We use CIFAR-100 samples and plot layer-wise gradient norm for different models and batch sizes. As shown, Instance Normalization [22] undergoes highest gradient explosion, followed by BatchNorm [1], GroupNorm [3], and LayerNorm [2] in all settings.

**Gradient Explosion in Other Normalizers:** We now extend the phenomenon of gradient explosion to other normalizers. The primary idea is that since all activation-based normalizers have a gradient expression similar to BatchNorm (i.e., projection followed by division by standard deviation), they all re-scale the gradient norm during backprop. However, the statistic used for normalization varies across normalizers, resulting in different severity of gradient explosion.

**Claim 4.** *For a given set of pre-activations, the backpropagated gradient undergoes higher average amplification through an Instance Normalization layer [22] than through a BatchNorm layer [1]. Further, GroupNorm [3] witnesses lesser gradient explosion than both these layers.*

*Proof.* The gradient backpropagated through the $g^{\text{th}}$ group in a GroupNorm layer with group-size $G$ is expressed as: $\nabla_{\mathbf{Y}_{L-1}^g}(J) = \frac{\gamma}{\sigma_{\{g\}}(X_L^g)} \mathcal{W}_L^T \mathcal{P}\left[\nabla_{\hat{\mathbf{X}}_L^g}(J)\right]$ (see appendix for derivation). Here, $\mathcal{P}$ is defined as: $\mathcal{P}[\mathbf{Z}] = \mathcal{P}_{\mathbf{1}}^{\perp}[\mathcal{P}_{\mathbb{S}(\hat{\mathbf{x}}_{L/\sqrt{G}})}^{\perp}[\mathbf{Z}]]$, where $\mathcal{P}_{\mathbb{S}(\hat{\mathbf{x}}_{L/\sqrt{G}})}^{\perp}[\mathbf{Z}] = (I - \frac{1}{G}\hat{\mathbf{X}}_L^g\hat{\mathbf{X}}_L^{g\,T})\mathbf{Z}$. That is, the component of gradient inline with the normalized pre-activations will be removed via projection onto the spherical manifold defined by $\|\hat{\mathbf{X}}_L^g\| = \sqrt{G}$. As can be seen, the gradient expressions for GroupNorm and BatchNorm are very similar. Hence, the discussion for gradient explosion in BatchNorm directly applies to GroupNorm as well. This implies, when Instance Normalization is used in a CNN, the gradient norm for a given channel $c$ and the $i^{\text{th}}$ sample is amplified by the factor $\frac{1}{\sigma_{\{x\}}(\mathbf{X}_{L,i}^c)}$ (inverse of spatial standard deviation). Then, over $N$ samples, using the arithmetic-mean $\geq$ harmonic-mean inequality, we see the average gradient amplification in Instance Normalization is greater than gradient amplification in BatchNorm: $\frac{1}{N}\sum_i \frac{1}{\sigma_{\{x\}}^2(\mathbf{X}_{L,i}^c)} \geq \frac{N}{\sum_i \sigma_{\{x\}}^2(\mathbf{X}_{L,i}^c)} = \frac{1}{\sigma_{\{N\}}^2(\mathbf{X}_L)}$.

Similarly applying arithmetic-mean $\geq$ harmonic-mean for a given sample and the $g^{\text{th}}$ group, we see average gradient amplification in Instance Normalization is greater than gradient amplification in GroupNorm: $\frac{1}{G}\sum_c \frac{1}{\sigma_{\{x\}}^2(\mathbf{X}_L^{g,c})} \geq \frac{G}{\sum_c \sigma_{\{x\}}^2(\mathbf{X}_L^{g,c})} = \frac{1}{\sigma_{\{g\}}^2(\mathbf{X}_L)}$. Extending this last inequality by averaging over $N$ samples, we see average gradient amplification in GroupNorm is lower than that in BatchNorm. *This implies grouping of neurons in GroupNorm helps reduce gradient explosion.* $\square$

We show empirical verification of Claim 4 in Figure 8. As can be seen, the gradient norm in earlier layers follows the order Instance Normaliation $\geq$ BatchNorm $\geq$ GroupNorm $\geq$ LayerNorm, as proved in Claim 4. Further, since increasing depth implies more normalization operations, we see gradient explosion increases as depth increases. Similarly, since reducing batch-size increases gradient noise, we find gradient explosion increases with decrease in batch-size as well.

**Speed–stability trade-off in GroupNorm:** Combined with Section 4, our discussion in this section helps identify a speed–stability trade-off in GroupNorm. Specifically, we find that while GroupNorm with group size equal to 1 (viz., Instance Normalization) results in more diverse features (see Claim 3), it is also more susceptible to gradient explosion and hence sees training instability for small batch-sizes/large model depth (see Figure 1). Meanwhile, when group size is equal to layer-width (viz., LayerNorm), gradient explosion can be avoided, but the model is unable to generate informative activations and thus witnesses slower optimization. Combining these

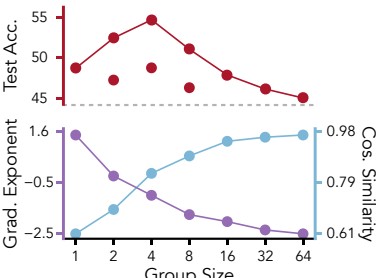

Figure 9: **Speed–Stability trade-off in GroupNorm** using 20-layer CNNs with batch-size 256 on CIFAR-100. We see increasing group size decreases gradient explosion (improved training stability) at the expense of high activation similarity (reduced optimization speed).

results demonstrates that the group size in GroupNorm ensues a trade-off between high similarity of activations (influences training speed) and gradient explosion (influences training stability). To illustrate this trade-off, we can estimate training instability by fitting an exponential curve to layerwise gradient norms (measures degree of gradient explosion) and estimate training speed by calculating cosine similarity of activations at the penultimate layer at initialization (highly correlated with training speed; see Figure 6). Results are shown in Figure 9. We see increasing group size clearly trades-off the two properties related to training speed and stability, with a moderately large group size resulting in best performance. In fact, we see test accuracy is highest exactly at this point of intersection in the trade-off. This explains the success of channel grouping in GroupNorm and other successful batch-independent normalization layers like EvoNormSO [27]. Interestingly, these results also help explain why in comparison to BatchNorm, which suffers from gradient explosion and exacerbates the problem of high gradient variance in non-IID Federated learning setups [52, 53], use of GroupNorm with a properly tuned group-size helps achieve better performance [52, 54].

## 6    Discussion and Limitations

**Discussion:** As the number of deep learning architectures continues to explode, the use of normalization layers is becoming increasingly common. However, past works provide minimal insight into what makes normalization layers beyond BatchNorm (un)successful. Our work acts as a starting point to bridge this gap. Specifically, we extend known results on benefits/limitations of BatchNorm to recently proposed normalization layers and provide a thorough characterization of their behavior at initialization. This generalized analysis provides a compass that can help systematically infer which normalization layer is most appealing under the constraints imposed by a given application, reducing reliance on empirical benchmarking. Moreover, since our results show phenomenon used to explain BatchNorm's success exist in alternative normalizers as well, we argue the success of BatchNorm requires further characterization. Our work also opens avenues for several new fronts of research. For example, in Section 4 we demonstrated that a normalization layer's impact on similarity of activations accurately predicts resulting optimization speed. As shown in a contemporary work by Boopathy and Fiete [55], the weight update dynamics of a neural network are in fact guided by the matrix defining similarity of activations. Beyond providing grounding to our observation, their results indicate that relating design choices in neural network development with similarity of activations can help optimize their values. Indeed, a recent method for neural architecture search directly utilizes the similarity of activations to design "good" architectures [56].

**Limitations:** In this work, we limit our focus to discriminative vision applications. We highlight that nine out of ten normalizers studied in this work were specifically designed for this setting and we indeed find that all our analyzed properties show predictive control over the final performance of a model in discriminative vision tasks, generalizing across multiple network architectures. However, extending our work to develop similar analyses in other contexts such as NLP will be very useful. The primary hurdle is that for different data modalities, the standard architecture families and their corresponding optimization difficulties vary widely. For example, in both LSTM and transformer architectures, an often noted training difficulty arises from large gradient norms, which can result in divergent training or training restarts [57, 58]. In fact, optimizers in existing NLP frameworks have gradient clipping enabled by default to avoid this problem [59]. Beyond large gradients, unbalanced gradients are also known to be a training difficulty in NLP architectures [60]. We think a thorough treatment of the role of normalization layers in addressing these problems will be very valuable and leave it for future work.

## Acknowledgements

We thank Hadi Daneshmand and anonymous reviewers for several helpful discussions that helped improve this paper. This work was partly supported by NSF under award CNS-2008151.

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
