# Appendix

The appendix is organized as follows:

## A  Experimental Configurations

Our experiments our implemented using PyTorch and code is available at `https://github.com/EkdeepSLubana/BeyondBatchNorm`. Throughout the paper, we record several useful properties like variance of activations or gradient norm w.r.t. activations. For this purpose, we use PyTorch's autograd library and define modules to calculate these properties without altering the input or gradients. Other experimental details are mentioned below.

**Model Architectures:** We focus on three model architectures: 10-layer non-residual networks, 20-layer non-residual networks, and ResNet-56. The non-residual networks architectures are detailed below. An integer $X$ represents the number of filters in the layer; the tuple $(X, y)$ represents the number of filters in the layer and the convolution's stride respectively; the function block(Z) represents a BasicBlock [24] used in ResNets, with X filters per layer and Z blocks. The final convolution layer's output is average-pooled and fed into a linear-classifier.

1. 10-layer non-residual networks: [64, (64, 2), 128, (128, 2), 256, (256, 2), 512, (512, 2), 512, 512]
2. 20-layer non-residual networks: [64, 64, 64, (64, 2), 128, 128, 128, (128, 2), 256, 256, 256, (256, 2), 256, 256, 256, (256, 2), 512, 512, 512, 512]
3. ResNet-56: [32, block(32, 9), block(64, 9), block(128, 9)]

**GroupNorm Configuration:** An important architectural detail is the number of groups in Group-Norm. Following standard configurations in past work [3], we define number of groups at any layer to be 32. In experiments where group size, instead of number of groups was fixed, the number of groups at any given layer are determined by layer-width divided by group-size. This ensures the group size stays constant across all layers.

**Batch-Sizes / Number of Training Epochs:** We used batch-sizes of 256 and 16 in this work. Models with batch-size 256 are trained for 60 epochs, while models with batch-size 16 are trained for 10 epochs. Note that an epoch of training with batch-size 16 corresponds to $50000/16 = 3125$ training iterations, while an epoch of training with batch-size 256 corresponds to $50000/256 \approx 256$ training iterations. Hence, even though we use fewer number of training epochs for batch-size 16, the overall training budget for batch-size 16 is approximately 2.5 times the training budget for training at batch-size 256.

**Learning Rates:** The initial learning rates are equal to either $0.1$ or $0.01$. Following Goyal et al. [21], we linearly scale the learning rate when changing batch-size. At 80% training progress, we decay the learning rate by a factor of 10. For models trained in Figure 5, the initial learning rate was set to ensure most normalizers can train a model effectively. Thus, for non-residual networks with 10 layers and ResNet-56, we use initial learning rate of $0.1$; for non-residual networks with 20 layers, we use initial learning rate of $0.01$.

**Other Hyperparameters:** We train using SGD and use weight decay of 0.0001 with momentum of 0.9 for all models. Note that weight decay is known to have an interesting interplay with BatchNorm during training [12, 61], however we leave the study of these effects for other normalizers for future work.

**Dataset / Preprocessing:** For training models in Figure 1, Figure 4, and Figure 5, we use CIFAR-100 with random horizontal flipping while training, but no other data augmentation is applied. The samples are pre-processed to have a mean/std of 0.5 in all dimensions, following standard practice with CIFAR datasets.

**Calculating Activations' Variance/Cosine Similarity/Gradient Norm:** In Figure 2, Figure 3, Figure 10, Figure 5, and Figure 8, we directly use CIFAR-100 samples without any augmentations for calculating the required properties. Samples are pre-processed to have a mean/std of 0.5 in all dimensions. We calculate the activations' variance/cosine similarity/gradient norm using custom modules (defined using PyTorch's autograd library) during forward/backward pass. All experiments use a batch-size of 256 and are averaged over 10 batches. For gradient explosion experiments using batch-size of 16, we average over 160 batches to match the number of samples used with batch-size 256.

**Calculating Rank/Cosine Similarity in Figure 6a and Figure 6b:** For calculating both rank and cosine similarity, we use random gaussian inputs. Via this, one can ensure the input has sufficiently distinct samples. Thus, if a normalizer results in high similarity of activations, that is an evident failure mode. This setup is in fact similar to Daneshmand et al. [4], who also use gaussian inputs for rank estimation. Following the authors, we use stable rank $\left( = \frac{\text{trace}(\Delta^T \Delta)}{\|\Delta\|_2^2} \text{ for a matrix } \Delta \right)$ for approximating rank in our experiments. This helps alleviate numerical precision problems, where relatively small magnitude singular values may end up contributing to the estimated rank.

# B   Gradient Derivations

We first redefine notations to focus on only the relevant variables. Specifically, we denote activations for a batch of $N$ samples at layer $L$ as $\mathbf{Y}_L \in \mathbb{R}^{D_L \times N}$, where $D_L$ is the dimension of a given sample's activations. The weight matrix is denoted as $\mathbf{W}_L \in \mathbb{R}^{D_{L-1} \times D_L}$. Two sets of outputs are relevant to us: (i) Pre-activations after weight multiplication, i.e., $\mathbf{X}_L = \mathbf{W}_L \mathbf{Y}_{L-1}$; (ii) Normalized pre-activations, i.e., $\hat{\mathbf{X}} = \mathcal{N}(\mathbf{X}_L)$, where $\mathcal{N}$ denotes a normalizer. We will derive gradients for both BatchNorm and GroupNorm to show similarity between their expressions.

## B.1   BatchNorm Gradients

Our derivation is similar to that of Chiley et al. [62], who also decompose the BatchNorm gradient into multiple projection operations. However, in that work, the authors focus on deriving an individual sample's gradient only; meanwhile, we derive gradients for the entire batch. This is important because when using BatchNorm, the gradient of a sample is affected by activations of a different sample ($i^{\text{th}}$ sample's activations use $j^{\text{th}}$ sample's activations for calculating mean and variance in normalization). We define the BatchNorm operation as follows [2]:

$$\begin{aligned} \hat{\mathbf{X}} &= \text{BN}(\mathbf{X}_L) \\ &= \sigma(\mathbf{X}_L)^{-1} \left( \mathbf{X}_L - \mu(\mathbf{X}_L) \right). \end{aligned} \tag{1}$$

Here, we define $\mu(\mathbf{X}_L) := \left( \frac{1}{N} \mathbf{X}_L \mathbf{1} \right) \mathbf{1}^T$ to perform mean-centering along the batch dimension and define $\sigma(\mathbf{X}_L) := \text{diag} \left( \left( \mathbf{X}_L - \left( \frac{1}{N} \mathbf{X}_L \mathbf{1} \right) \mathbf{1}^T \right) \left( \mathbf{X}_L - \left( \frac{1}{N} \mathbf{X}_L \mathbf{1} \right) \mathbf{1}^T \right)^T \right)^{1/2}$ to perform the variance normalization operation. This method of defining BatchNorm is similar to that of Daneshmand et al. [4].

Note that the $k^{\text{th}}$ diagonal element in matrix $\sigma(\mathbf{X}_L)$, denoted $\sigma(\mathbf{X}_L)^{(k)}$, is the variance along the batch-dimension for the $k^{\text{th}}$ index across all samples' pre-activations. We now calculate the derivative of this element with respect to the $i^{\text{th}}$ sample's pre-activations, i.e., $\mathbf{X}_{L,i}$. Since only the $k^{\text{th}}$ element

---

[2]We ignore the affine scale and shift parameters (defined as $\gamma, \beta$ in the main paper) as they are initialized to one and zero, respectively. That is, they do not influence our discussion in any way at initialization.

of $\mathbf{X}_{L,i}$, denoted $\mathbf{X}_{L,i}^{(k)}$, is of relevance to the variance calculation operation, we focus on it specifically. We use $\mu(\mathbf{X}_L)^{(k)}$ to index the mean matrix. This leads to the following:

$$\sigma(\mathbf{X}_L)^{(k)} = \left( \frac{1}{N} \sum_n \left( \mathbf{X}_{L,n}^{(k)} - \mu(\mathbf{X}_L)^{(k)} \right)^2 \right)^{1/2}.$$

$$\implies \frac{\partial}{\partial \mathbf{X}_{L,i}^{(k)}} \sigma(\mathbf{X}_L)^{(k)} = \frac{1}{N \, \sigma(\mathbf{X}_{L,i})^{(k)}} \sum_n \left( \left( \mathbf{X}_{L,n}^{(k)} - \mu(\mathbf{X}_L)^{(k)} \right) \left( \delta_{i,n} - \frac{1}{N} \right) \right)$$

$$= \frac{1}{N \, \sigma(\mathbf{X}_{L,i})^{(k)}} \left( \mathbf{X}_{L,i}^{(k)} - \mu(\mathbf{X}_L)^{(k)} \right)$$

$$= \frac{1}{N} \hat{\mathbf{X}}_{L,i}^{(k)}$$

(2)

where $\delta_{i,n} = 1$ if $i = n$ and 0 otherwise. A similar calculation can be conducted for the mean matrix too:

$$\mu(\mathbf{X}_L)^{(k)} = \frac{1}{N} \sum_n \left( \mathbf{X}_{L,n}^{(k)} \right) \implies \frac{\partial}{\partial \mathbf{X}_{L,i}^{(k)}} \mu(\mathbf{X}_L)^{(k)} = \frac{1}{N}.$$

(3)

Now, for the $j^{\text{th}}$ sample's normalized pre-activations, we have the following:

$$\frac{\partial}{\partial \mathbf{X}_{L,i}^{(k)}} \hat{\mathbf{X}}_{L,j}^{(k)} = \frac{\partial}{\partial \mathbf{X}_{L,i}^{(k)}} \left( \frac{\mathbf{X}_{L,j}^{(k)} - \mu(\mathbf{X}_L)^{(k)}}{\sigma(\mathbf{X}_L)^{(k)}} \right)$$

$$= \frac{1}{(\sigma(\mathbf{X}_L)^{(k)})^2} \left[ \sigma(\mathbf{X}_L)^{(k)} \left( \delta_{i,j} - \frac{\partial}{\partial \mathbf{X}_{L,i}^{(k)}} \mu(\mathbf{X}_L)^{(k)} \right) - \left( \mathbf{X}_{L,j}^{(k)} - \mu(\mathbf{X}_L)^{(k)} \right) \frac{\partial}{\partial \mathbf{X}_{L,i}^{(k)}} \sigma(\mathbf{X}_L)^{(k)} \right].$$

We can use expressions derived in Equations 2 and 3 to complete the above equation as follows:

$$\frac{\partial}{\partial \mathbf{X}_{L,i}^{(k)}} \hat{\mathbf{X}}_{L,j}^{(k)} = \frac{1}{\sigma(\mathbf{X}_L)^{(k)}} \left( \left( \delta_{i,j} - \frac{1}{N} \right) - \frac{1}{N} \hat{\mathbf{X}}_{L,i}^{(k)} \hat{\mathbf{X}}_{L,j}^{(k)} \right).$$

(4)

We now calculate the gradient of loss ($J$) with respect to $\mathbf{X}_{L,i}^{(k)}$. Specifically, we use the relationship in Equation 4 to derive the following:

$$\frac{\partial}{\partial \mathbf{X}_{L,i}^{(k)}} (J) = \sum_j \frac{\partial}{\partial \hat{\mathbf{X}}_{L,j}^{(k)}} (J) \frac{\partial}{\partial \mathbf{X}_{L,i}^{(k)}} \hat{\mathbf{X}}_{L,j}^{(k)}$$

$$= \sum_j \frac{\partial}{\partial \hat{\mathbf{X}}_{L,j}^{(k)}} (J) \left( \frac{1}{\sigma(\mathbf{X}_L)^{(k)}} \left( \left( \delta_{i,j} - \frac{1}{N} \right) - \frac{1}{N} \hat{\mathbf{X}}_{L,i}^{(k)} \hat{\mathbf{X}}_{L,j}^{(k)} \right) \right)$$

$$= \frac{1}{\sigma(\mathbf{X}_L)^{(k)}} \left( \frac{\partial}{\partial \hat{\mathbf{X}}_{L,i}^{(k)}} (J) - \frac{1}{N} \sum_j \frac{\partial}{\partial \hat{\mathbf{X}}_{L,j}^{(k)}} (J) - \frac{\hat{\mathbf{X}}_{L,i}^{(k)}}{N} \sum_j \hat{\mathbf{X}}_{L,j}^{(k)} \frac{\partial}{\partial \hat{\mathbf{X}}_{L,j}^{(k)}} (J) \right)$$

$$= \frac{1}{\sigma(\mathbf{X}_L)^{(k)}} \left( \frac{\partial}{\partial \hat{\mathbf{X}}_{L,i}^{(k)}} (J) - \frac{1}{N} \nabla_{\hat{\mathbf{X}}_L} (J)^{(k)} \mathbf{1} - \frac{\hat{\mathbf{X}}_{L,i}^{(k)}}{N} \nabla_{\hat{\mathbf{X}}_L} (J)^{(k)} (\hat{\mathbf{X}}_L^{(k)})^T \right).$$

(5)

Here, $\nabla_{\hat{\mathbf{X}}_L} (J)^{(k)}$ denotes the $k^{\text{th}}$ row of the gradient of loss with respect to normalized pre-activations $\hat{\mathbf{X}}_L$. Similarly, $\hat{\mathbf{X}}_L^{(k)}$ denotes the $k^{\text{th}}$ row of the normalized pre-activations matrix.

We can extend Equation 5 to the entire pre-activations matrix, as shown in the following:

$$\nabla_{\mathbf{X}_L} (J) = \sigma(\mathbf{X}_L)^{-1} \left( \nabla_{\hat{\mathbf{X}}_L} (J) - \frac{1}{N} \nabla_{\hat{\mathbf{X}}_L} (J) \mathbf{1} \mathbf{1}^T - \frac{1}{N} \text{diag} \left( \nabla_{\hat{\mathbf{X}}_L} (J) \hat{\mathbf{X}}_L^T \right) \hat{\mathbf{X}}_L \right).$$

(6)

Define the Oblique-manifold $\text{Ob}(\frac{1}{\sqrt{N}} \hat{\mathbf{X}}_L) := \frac{1}{N} \text{diag}(\hat{\mathbf{X}}_L \hat{\mathbf{X}}_L^T) = \text{diag}(\mathbf{1})$ and the projection operator $\mathcal{P}^{\perp}_{\text{Ob}(\frac{1}{\sqrt{N}} \hat{\mathbf{X}}_L)}[\mathbf{Z}] = \mathbf{Z} - \frac{1}{N} \text{diag}(\mathbf{Z} \, \hat{\mathbf{X}}_L^T) \hat{\mathbf{X}}_L$ that removes the component of $Z$ inline with the normalized pre-activations. Similarly, define the projection operator related to the all-ones vector $\mathbf{1}$, which mean-centers its input along the batch-dimension: $\mathcal{P}^{\perp}_{\mathbf{1}}[\mathbf{Z}] = \mathbf{Z} \left( I - \frac{1}{N} \mathbf{1} \mathbf{1}^T \right)$. Using these definitions along

with the fact the normalized pre-activations satisfy $\hat{\mathbf{X}}_\mathbf{L}\mathbf{1} = \mathbf{0}$, we get:

$$\nabla_{\mathbf{X}_L}(J) = \sigma(\mathbf{X}_L)^{-1} \left( \mathcal{P}_\mathbf{1}^\perp \left[ \mathcal{P}_{\text{Ob}(\frac{1}{\sqrt{N}}\hat{\mathbf{x}}_L)}^\perp \left[ \nabla_{\hat{\mathbf{x}}_L}(J) \right] \right] \right)$$

$$= \sigma(\mathbf{X}_L)^{-1} \left( \mathcal{P} \left[ \nabla_{\hat{\mathbf{x}}_L}(J) \right] \right), \tag{7}$$

where, we used an overall projection operation $\mathcal{P}[.] = \mathcal{P}_\mathbf{1}^\perp \left[ \mathcal{P}_{\text{Ob}(\frac{1}{\sqrt{N}}\hat{\mathbf{x}}_L)}^\perp [.] \right]$.

Now, recall $\mathbf{X}_L = \mathbf{W}_L\mathbf{Y}_{L-1}$. Thus, $\nabla_{\mathbf{Y}_{L-1}}(J) = \mathbf{W}_L^T\nabla_{\mathbf{X}_L}(J)$. We now redefine $\sigma$ in the general way used in the main paper, i.e., $\sigma_{\{N\}}(\mathbf{X}_L)$, to denote the batch-variance operator that calculates variance along the batch-dimension $\{N\}$. Then, using Equation 7, we get our desired result:

$$\nabla_{\mathbf{Y}_{L-1}}(J) = \frac{1}{\sigma_{\{N\}}(\mathbf{X}_L)} \mathbf{W}_L^T \left( \mathcal{P} \left[ \nabla_{\hat{\mathbf{x}}_L}(J) \right] \right), \tag{8}$$

as was shown in the main paper. This completes the derivation.

## B.2 GroupNorm Gradients

We now derive gradients for GroupNorm. The derivation is quite similar to that of BatchNorm, but the definitions and order of operations changes substantially now. As we will see though, the final expression is essentially the same: a projection operation followed by a division by variance operation.

To begin, we first note that unlike BatchNorm, a given sample is normalized independently of other samples and hence we can analyze only a single sample for our discussion. We thus analyze activations $\mathbf{Y}_L \in \mathbb{R}^{D_L \times 1}$ for a single sample at layer $L$, distinguishing between pre-activations ($\mathbf{X}_L = \mathbf{W}\mathbf{Y}_{L-1}$) and normalized pre-activations ($\hat{\mathbf{X}}_L = \mathcal{N}(\mathbf{X}_L)$), as we did before in BatchNorm. We define the GroupNorm operation as follows[3]:

$$\hat{\mathbf{X}}^{(g)} = \text{GN}(\mathbf{X}_L^{(g)})$$

$$= \frac{1}{\sigma_g} \left( \mathbf{X}_L^{(g)} - \mu_g\mathbf{1} \right). \tag{9}$$

Here, we assume the activations $\mathbf{X}_L$ have been divided into groups of size $G$ and we use the notation $\mathbf{X}_L^{(g)}$ to denote the $g^{\text{th}}$ group. The variable $\mu_g$ denotes the mean of the $g^{\text{th}}$ group's activations and $\sigma_g$ denotes the standard deviation. That is, $\mu_g = \frac{1}{G}\mathbf{X}_L^{(g)T}\mathbf{1}$, where $\mathbf{1}$ is an all-ones vector of dimension $G$. Similarly, $\sigma_g = \sqrt{\frac{1}{G}\sum_k \left( \mathbf{X}_{L,k}^{(g)} - \mu_g \right)^2}$, where, *in contrast with our discussion on BatchNorm, we use $X_{L,k}$ to index the neuron or channel in a given sample $\mathbf{X}_L$*.

With the notations established, we now begin the main derivation. We first calculate the partial derivative of the mean and standard deviation with respect to $\mathbf{X}_{L,i}^{(g)}$.

$$\sigma_g = \left( \frac{1}{G}\sum_k \left( \mathbf{X}_{L,k}^{(g)} - \mu_g \right)^2 \right)^{1/2}.$$

$$\implies \frac{\partial}{\partial \mathbf{X}_{L,i}^{(g)}} \sigma_g = \frac{1}{G\sigma_g}\sum_k \left( \left( \mathbf{X}_{L,k}^{(g)} - \mu_g \right) \left( \delta_{i,k} - \frac{1}{G} \right) \right)$$

$$= \frac{1}{G\sigma_g} \left( \mathbf{X}_{L,i}^{(g)} - \mu_g \right)$$

$$= \frac{1}{G}\hat{\mathbf{X}}_{L,i}^{(g)} \tag{10}$$

where $\delta_{i,n} = 1$ if $i = n$ and 0 otherwise. A similar calculation can be conducted for the mean matrix too:

$$\mu_g = \frac{1}{G}\sum_k \left( \mathbf{X}_{L,k}^{(g)} \right) \implies \frac{\partial}{\partial \mathbf{X}_{L,i}^{(g)}} \mu_g = \frac{1}{G}. \tag{11}$$

---

[3]We again ignore the affine scale and shift parameters (defined as $\gamma$, $\beta$ in the main paper) as they are initialized to one and zero, respectively. That is, they do not influence our discussion in any way at initialization.

Since the $i^{\text{th}}$ neuron (or channel) plays a role in the normalization of the $j^{\text{th}}$ sample's pre-activations through mean/variance estimation operations, they will influence each other's gradients during the backward pass. Thus, we have the following:

$$\frac{\partial}{\partial \mathbf{X}_{L,i}^{(g)}} \hat{\mathbf{X}}_{L,i}^{(g)} = \frac{\partial}{\partial \mathbf{X}_{L,i}^{(g)}} \left( \frac{\mathbf{X}_{L,j}^{(g)} - \mu_{(g)}}{\sigma_g} \right)$$

$$= \frac{1}{(\sigma_g)^2} \left[ \sigma_g \left( \delta_{i,j} - \frac{\partial}{\partial \mathbf{X}_{L,i}^{(g)}} \mu_g \right) - \left( \mathbf{X}_{L,j}^{(g)} - \mu_g \right) \frac{\partial}{\partial \mathbf{X}_{L,i}^{(g)}} \sigma_g \right]$$

$$= \frac{1}{(\sigma_g)^2} \left[ \sigma_g \left( \delta_{i,j} - \frac{1}{G} \right) - \left( \mathbf{X}_{L,j}^{(g)} - \mu_g \right) \frac{1}{G} \hat{\mathbf{X}}_{L,i}^{(g)} \right]$$

$$= \frac{1}{(\sigma_g)} \left[ \left( \delta_{i,j} - \frac{1}{G} \right) - \frac{1}{G} \hat{\mathbf{X}}_{L,i}^{(g)} \hat{\mathbf{X}}_{L,j}^{(g)} \right],$$

(12)

where we used expressions derived in Equations 10 and 11 to arrive at the second-last equality. We now calculate the gradient of loss ($J$) with respect to $\mathbf{X}_{L,i}^{(k)}$. Specifically, we use the relationship in Equation 12 to derive the following:

$$\frac{\partial}{\partial \mathbf{X}_{L,i}^{(g)}}(J) = \sum_j \frac{\partial}{\partial \hat{\mathbf{X}}_{L,j}^{(g)}}(J) \frac{\partial}{\partial \mathbf{X}_{L,i}^{(g)}} \hat{\mathbf{X}}_{L,j}^{(g)}$$

$$= \sum_j \frac{\partial}{\partial \hat{\mathbf{X}}_{L,j}^{(g)}}(J) \left( \frac{1}{\sigma_g} \left( \left( \delta_{i,j} - \frac{1}{G} \right) - \frac{1}{G} \hat{\mathbf{X}}_{L,i}^{(g)} \hat{\mathbf{X}}_{L,j}^{(g)} \right) \right)$$

$$= \frac{1}{\sigma_g} \left( \frac{\partial}{\partial \hat{\mathbf{X}}_{L,i}^{(g)}}(J) - \frac{1}{G} \sum_j \frac{\partial}{\partial \hat{\mathbf{X}}_{L,j}^{(g)}}(J) - \frac{\hat{\mathbf{X}}_{L,i}^{(g)}}{G} \sum_j \hat{\mathbf{X}}_{L,j}^{(g)} \frac{\partial}{\partial \hat{\mathbf{X}}_{L,j}^{(g)}}(J) \right)$$

$$= \frac{1}{\sigma_g} \left( \frac{\partial}{\partial \hat{\mathbf{X}}_{L,i}^{(g)}}(J) - \frac{1}{G} \mathbf{1}^T \nabla_{\hat{\mathbf{X}}_L^{(g)}}(J) - \frac{1}{G} \left( \hat{\mathbf{X}}_L^{(g)\,T} \nabla_{\hat{\mathbf{X}}_L^{(g)}}(J) \right) \hat{\mathbf{X}}_{L,i}^{(g)} \right).$$

(13)

Here, the vector $\nabla_{\hat{\mathbf{X}}_L^{(g)}}(J)$ denotes the gradient of the loss with respect to the $g^{\text{th}}$ group of normalized pre-activations $\hat{\mathbf{X}}_L^{(g)}$. The above equation is with respect to an individual element of the pre-activations, so we now expand it to the entire group as follows:

$$\nabla_{\mathbf{X}_L^{(g)}}(J) = \frac{1}{\sigma_g} \left( \nabla_{\hat{\mathbf{X}}_L^{(g)}}(J) - \frac{1}{G} \left( \mathbf{1}^T \nabla_{\hat{\mathbf{X}}_L^{(g)}}(J) \right) \mathbf{1} - \frac{1}{G} \left( \hat{\mathbf{X}}_L^{(g)\,T} \nabla_{\hat{\mathbf{X}}_L^{(g)}}(J) \right) \hat{\mathbf{X}}_L^{(g)} \right)$$

$$= \frac{1}{\sigma_g} \left( I - \frac{1}{G} \mathbf{1}\mathbf{1}^T \right) \left( I - \frac{1}{G} \hat{\mathbf{X}}_L^{(g)} \hat{\mathbf{X}}_L^{(g)\,T} \right) \nabla_{\hat{\mathbf{X}}_L^{(g)}}(J)$$

$$= \frac{1}{\sigma_g} \mathcal{P}_{\mathbf{1}}^{\perp} \left[ \mathcal{P}_{\mathbb{S}(\hat{\mathbf{X}}_L^{(g)})}^{\perp} \left[ \nabla_{\hat{\mathbf{X}}_L^{(g)}}(J) \right] \right]$$

$$= \frac{1}{\sigma_g} \mathcal{P} \left[ \nabla_{\hat{\mathbf{X}}_L^{(g)}}(J) \right].$$

(14)

In the above, we defined the projection operator $\mathcal{P}[.]$ to compose two separate projection operators. Specifically, the operator $\mathcal{P}_{\mathbf{1}}^{\perp}[\mathbf{Z}] = \left( I - \frac{1}{G} \mathbf{1}\mathbf{1}^T \right) \mathbf{Z}$, which is similar as defined before in BatchNorm's derivation, with the slight difference that uses right-hand matrix multiplication (the geometric interpretation remains the same, however). Meanwhile, the operator $\mathcal{P}_{\mathbb{S}(\hat{\mathbf{X}}_L^{(g)})}^{\perp}[\mathbf{Z}] = \left( I - \frac{1}{G} \hat{\mathbf{X}}_L^{(g)} \hat{\mathbf{X}}_L^{(g)\,T} \right) \mathbf{Z}$ removes the component of its input inline with the $g^{\text{th}}$ group of activations for the given sample by projecting it onto the spherical manifold defined by $\mathbb{S}(\hat{\mathbf{X}}_L^{(g)})$. This is similar to the Oblique-manifold projector in BatchNorm; however, this operator focus on an individual sample only.

Now, recall $\mathbf{X}_L = \mathbf{W}_L \mathbf{Y}_{L-1}$. Thus, $\nabla_{\mathbf{Y}_{L-1}}(J) = \mathbf{W}_L^T \nabla_{\mathbf{X}_L}(J)$. We now redefine $\sigma$ in the general way used in the main paper, i.e., $\sigma_{\{g\}}(\mathbf{X}_L)$, to denote the group-variance operator that calculates variance along the group-dimension $\{g\}$. Then, using Equation 14, we can complete the derivation:

$$\nabla_{\mathbf{Y}_{L-1}^{(g)}}(J) = \frac{1}{\sigma_{\{g\}}(\mathbf{X}_L)} \mathbf{W}_L^T \left( \mathcal{P} \left[ \nabla_{\hat{\mathbf{X}}_L^{(g)}}(J) \right] \right).$$

(15)

Comparing the final gradient expression for GroupNorm (Equation 15) and BatchNorm (Equation 8), we can clearly see both GroupNorm's and BatchNorm's gradient essentially involve a projection operation, followed by matrix multiplication with the layer weights and division by standard deviation of the pre-activations. The primary difference lies in how the standard deviation is computed. For GroupNorm, we use a group's standard deviation, which is bound to be small than a Batch's standard deviation (as argued in Section 5 using the arithmetic-mean $\geq$ harmonic-mean inequality). The results in Figure 8 provide empirical demonstration of this claim.

## C Circumventing Exploding/Vanishing Activations in Non-Residual Networks

For the case of non-residual networks, the problem of growing/vanishing variance does not manifest generally. To understand this, note that activations-based normalizers act on a combination of the batch, the channel, or the spatial dimensions and rescale the variance (or norm) in those dimensions to 1, thus avoiding both exploding/vanishing activations. For parametric normalizers, as discussed in Section 3.2, properly designed corrections can ensure the signal variance is preserved during forward propagation. This implies, by design, all normalizers mentioned in Table 1 ensure that activations neither explode nor vanish in the case of non-residual networks. We empirically demonstrate this behavior in Figure 10 for a 20-layer CNN. Except for Scaled Weight Standardization [23], all methods ensure the variance neither grows nor vanishes, hence corroborating our argument. Indeed, for a

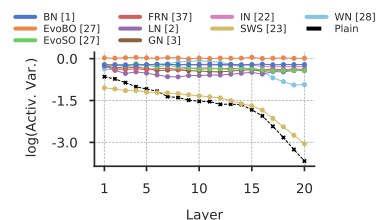

Figure 10: Log Activation Variance as a function of layer number in a 20 layer non-residual CNN. As shown, most normalization methods can avoid the problem of exploding/vanishing variance.

20-layer CNN, Scaled Weight Standardization is unable to train for most training configurations we tried (see Figure 12).

## D More Results

### D.1 Train/Test Curves for Different Configurations of Model Architecture, Normalizer, Batch-Size, and Initial Learning Rate

In the main paper, we provided training progress curves for a few configurations of model architecture, normalizer, batch-size, and initial learning rate. Here, we provide both train/test curves and several more settings. Specifically, Figure 11 contains results for non-residual CNNs with 10 layers; Figure 12 contains results for non-residual CNNs with 20 layers; Figure 13 contains results on ResNet-56 without SkipInit [6]; and Figure 14 contains results on ResNet-56 with SkipInit [6]. All results are averaged over 3 seeds and error bars show $\pm$ standard deviation.

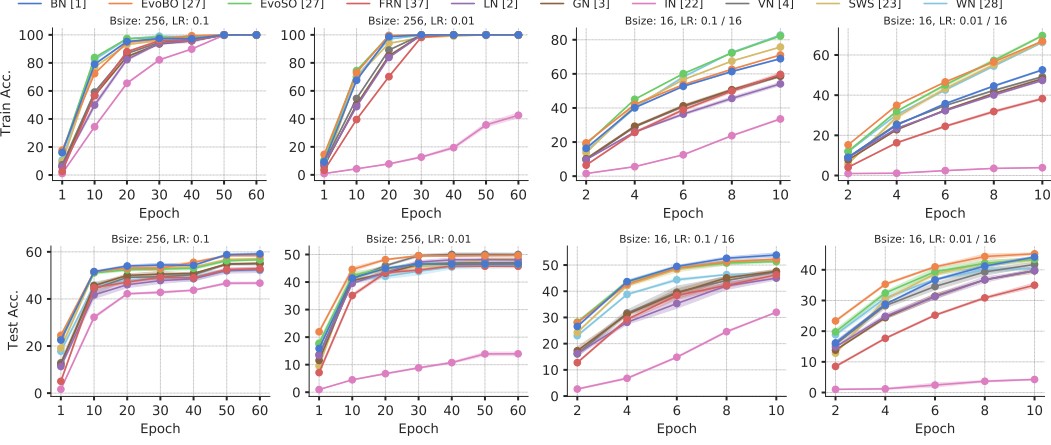

Figure 11: Non-Residual CNN with 10 layers

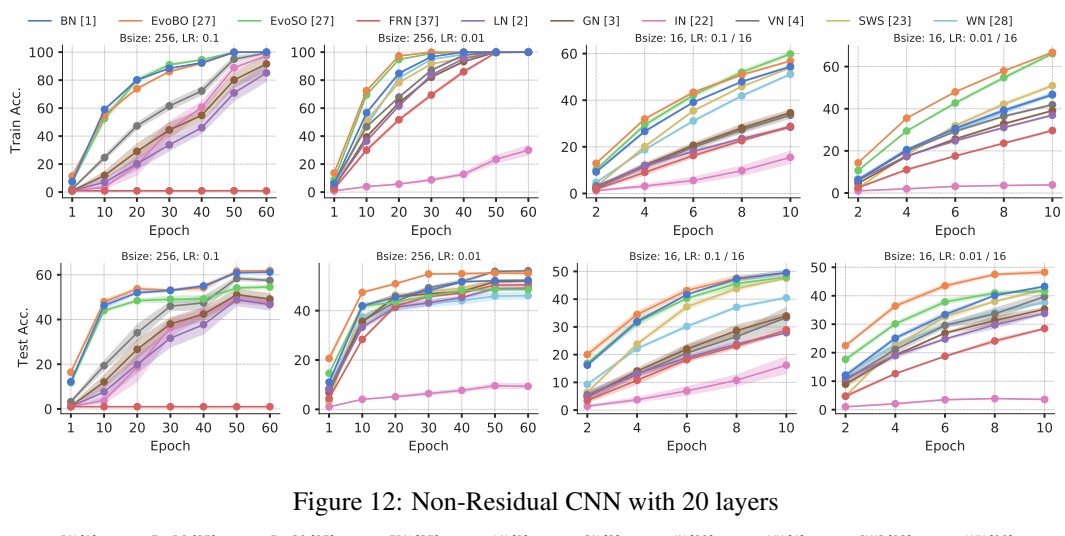

Figure 12: Non-Residual CNN with 20 layers

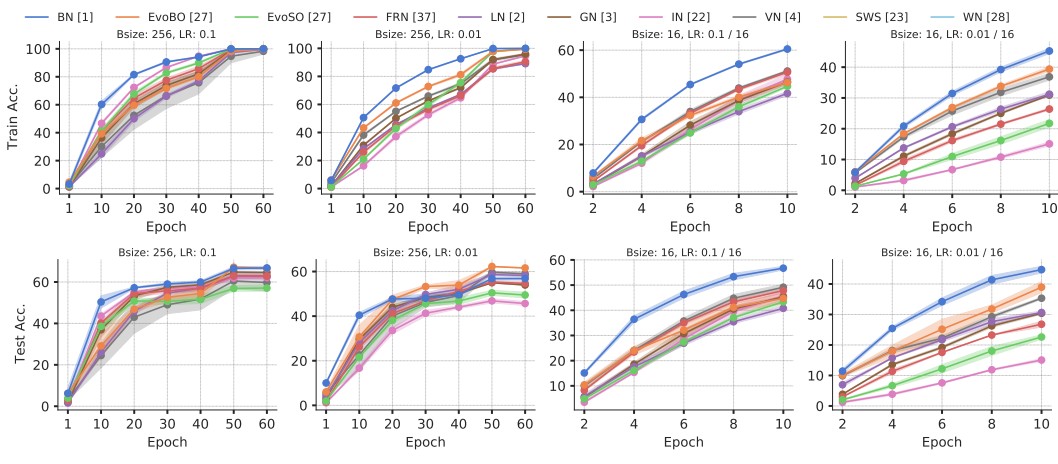

Figure 13: ResNet-56 without SkipInit

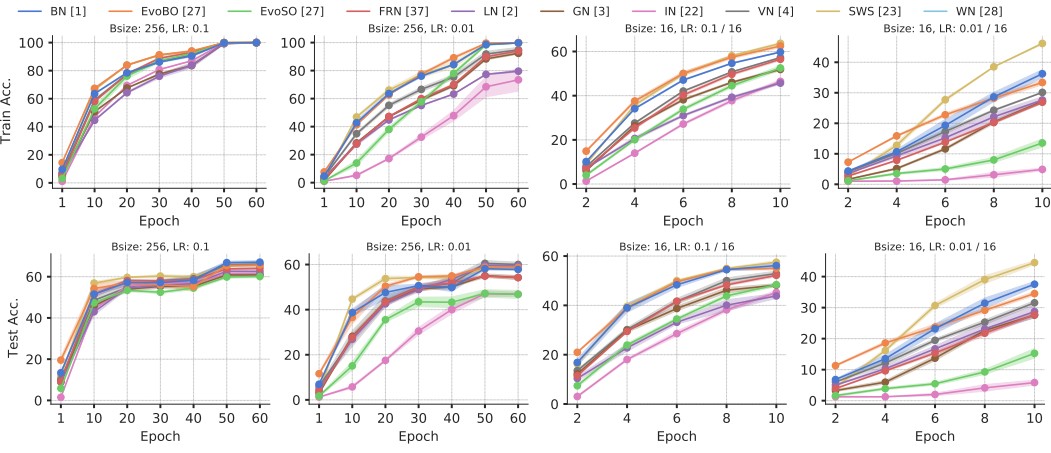

Figure 14: ResNet-56 with SkipInit

## D.2 Cosine Similarity for Different Normalizers and Model Architecture Combinations

In the main paper, we discussed the implications of high similarity of activations for different input samples. Based on our discussion, we showed the tradeoff between group-size and cosine similarity

of activations in GroupNorm (and hence Instance Normalization/LayerNorm). However, it is worth noting that except for EvoNormSO [27], all normalizers use either batch variance or group variance for normalization. This indicates our discussion was general and hence applicable to all layers discussed in this paper. We show demonstration of this argument in Figure 15, where we find layers that rely on batch or group variance normalization have lower cosine similarity of activations.

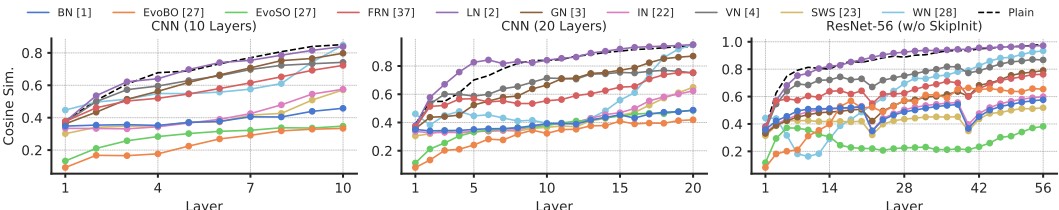

Figure 15: Cosine Similarity of activations for Non-Residual CNN with 10 layers, 20 layers, and ResNet-56 without SkipInit.

## D.3 Gradient Explosion for Different Normalizers and Model Architecture Combinations

In Section 5, we discussed how gradient explosion instantiates in normalization layers which rely on batch-statistics or instance-statistics, while layers which use group-statistics can avoid this behavior. Here, we demonstrate our discussion holds generally true. For example, in Figure 16, we show gradient explosion is highest in Instance Normalization [22] and follows the order Instance Normalization $\geq$ BatchNorm [1] $\geq$ GroupNorm [3] $\geq$ LayerNorm [2], as expected by our analysis, across several model architectures. Yang et al. [29] show ResNets are able to avoid gradient explosion by design. However, for the case of small batch-size (see Figure 17), we find gradient explosion continues to occur with Instance Normalization, even in ResNets. This again explains Instance Normalization's inability to train effectively and stably.

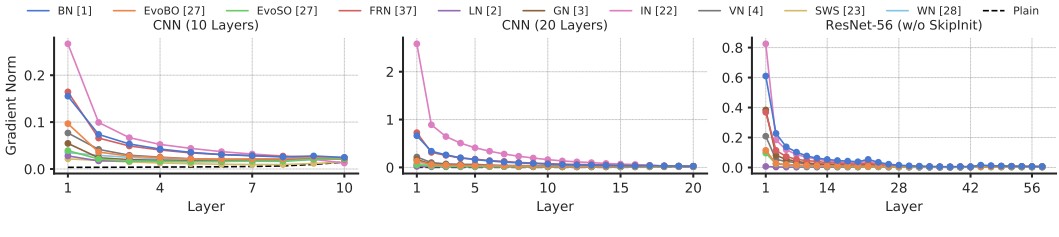

Figure 16: Batch-Size: 256

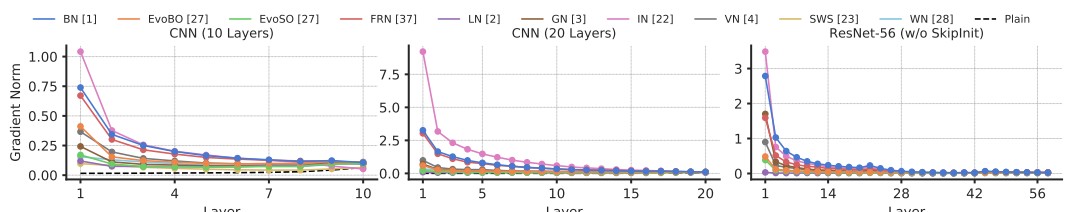

Figure 17: Batch-Size: 16