# OpenReview forum: "Beyond BatchNorm: Towards a Unified Understanding of Normalization in Deep Learning"
_NeurIPS.cc/2021/Conference — NeurIPS 2021 Poster_

### Official Review · Reviewer_gpva · 2021-07-15

**Rating:** 7
**Confidence:** 3

**Summary:**

This paper summarizes multiple aspects of a variety of normalization schemes and what makes them work or not work. Specifically, the authors analyze properties that prior works have found to be critical in BatchNormalso in the context of other normalizers (like GroupNorm), thus arguing for a more unified understanding and insight towards these normalization components. The discussion and thus contribution can be broken into three parts: stable activation, discriminative representation and gradient explosion analysis.

**Limitations And Societal Impact:**

I would indeed prefer that the authors expand upon the limitation of their analysis further (e.g., the LayerNorm and NLP point I mentioned above). There is no immediate societal impact that I think this paper should address.

**Main Review:**

Overall, I find the paper a very interesting read. The paper is not completely novel as all of the properties studied here have already been discussed in prior works (e.g., rank collapse, gradient explosion, stable activation), but the authors manage to extend these discussions to other sorts of activation-based normalizers and even parametric normalizers.

Pros:
- Clear explanation and intuition; the paper is well-written overall.
- The proof is correct (upon a cursory look) and relatively straightforward.
- Good set of empirical evidence supporting most of the claims and conjectures of the authors.
- Thorough examination of the multiple aspects of the normalization components and the conclusion is a good step towards further unifying these normalizers.

Cons:
- Relatively limited context in terms of the discussion (e.g., a major portion of this paper still focuses on activation-based normalizers, in particular BN, GN (and its variant IN); learnable parameter not discussed).
- While the unified perspective is appealing and with good empirical support, I'm dubious of whether the property claims can generalize to other contexts and domains (i.e., are they necessary? are they sufficient?). See my comments below.

--------------------------------
I have more questions/comments/feedbacks for the authors:

1. The first major concern I have over the analysis presented by the authors is the absence of analysis of the role learnable parameters play in this context. For example, with different layers having different $(\gamma, \beta)$, wouldn't the activation variance growth of a deep ResNet (e.g., Fig. 2) be more unstable? Wouldn't the gradient explosion derivation also have to include a $\gamma$ factor in Eq. (1)? Were the experiments run with or without the affine parameters?

2. The second major question is whether these properties are necessary/sufficient for building "good normalizers". For example, the analysis in the paper overall does not explain why LayerNorm is so indispensable in the NLP context (e.g., in Transformers). According to the analysis of the authors, LayerNorm yields similar representations and seems generally not preferred over GroupNorm (in fact, it is generally not a good idea to use BN in NLP tasks, I think). Can the authors comment more on this?

3. What is the reason for vanishing variance of SWS in Figure 5? With the correction scaling, wouldn't one expect its activation variance to be relatively stable?

4. This is more of a question of curiosity that I'd like to gather the authors' thoughts on: in the case where the parameters are shared across the residual blocks (i.e., $f_L = f_{L-1} = \dots = f_1$), how will the behavior of the normalization components change? I'm mainly thinking of the formulation of Neural ODE and other implicit models, which could also take the form $\mathbf{y}_L = \mathbf{y}_{L-1} + \mathcal{N}(f(\mathbf{y}_{L-1}))$ (written in a differential, forward Euler form), but have infinite layers. Obviously, their activations did not explode, and neither did their gradient.

5. Where is the correlation analysis of $\|\nabla_{Y_L}(J)\|$ vs. $\prod_{l=10}^{L+1} \sigma_N$ mentioned in L340-341? I didn't find it in the appendix.

-----------------------------

**Minor points**:

1. L302-303: Isn't the takeaway that smaller group size yield more discriminative features?
2. L319: from layer $L$ to layer $L-1$.
3. L341: $\sigma_N(X_l)$.
4. L594: $\mathbf{W}_L \in \mathbb{R}^{D_L \times D_{L-1}}$.
5. L606: You missed a $\frac{1}{N}$ in $\text{diag}$ in the expression of $\sigma(\mathbf{X}_L)$.

----------------
----------------
#### Post-rebuttal

I've read the authors' response and the other reviews. I generally agree with the other reviewers that there are still quite a few observations revealed in the paper that are not yet fully explained (even empirically), but I also acknowledge that this would be interesting for future work. I'm overall satisfied with the authors' response and am inclined to keep my score.

**Time Spent Reviewing:**

6

---

> ### Author Response · Authors · 2021-08-09
> **Response to Reviewer gpva (Part 1)**
>
> > **"The first major concern I have over the analysis presented by the authors is the absence of analysis of the role learnable parameters play in this context. For example, with different layers having different ($\gamma$, $\beta$), wouldn't the activation variance growth of a deep ResNet (e.g., Fig. 2) be more unstable? Wouldn't the gradient explosion derivation also have to include a γ factor in Eq. (1)? Were the experiments run with or without the affine parameters?"**
>
> We answer this question in two parts below.
>
> 1. Why affine parameters do not play a role in our analysis: We highlight to the reviewer that all properties studied in our work are analyzed *at initialization*. Across all machine learning frameworks, the affine scale ($\gamma$) and bias ($\beta$) parameters are initialized to ones and zeros respectively, and hence they do not affect any of our analyzed properties.
>
> 2. Beyond initialization: Affine parameters will change during training, however that does not affect our analysis. This is so because the *fatal* impact of our analyzed properties happens at initialization itself. For example, variance explosion at initialization leads to divergent training (loss goes to infinity), rendering the model un-trainable. Similarly, when gradient explosion is large (e.g., in the case of Instance Normalization), it again leads to divergent training at initialization itself, especially when used with large model depth and small batch-size. Finally, ensuring high rank of output representations helps ensure the model produces discriminative representations at initialization, essentially giving a jump-start to the training process (one can also think of this as an useful inductive bias for discriminative applications). Overall, since the positive/negative effects of our analyzed properties happen right at initialization, even though the affine parameters will change beyond initialization, they do not affect any of our conclusions.
>
> *Summary:* Since affine scale and bias parameters are initialized to ones and zeros, they do not affect the properties studied in our work. Further, since these properties affect training dynamics immediately at initialization, the values of affine parameters beyond initialization do not affect any of our conclusions. We will also make sure to better stress in the paper that affine parameters do not play a role in our analysis.
>
>
> > **"The second major question is whether these properties are necessary/sufficient for building 'good normalizers'. For example, the analysis in the paper overall does not explain why LayerNorm is so indispensable in the NLP context (e.g., in Transformers). According to the analysis of the authors, LayerNorm yields similar representations and seems generally not preferred over GroupNorm (in fact, it is generally not a good idea to use BN in NLP tasks, I think). Can the authors comment more on this?"**
>
> Our work's primary focus revolves around discriminative vision applications. We highlight that 9 out of 10 normalizers studied in our work were specifically designed for this setting and we indeed find that all our analyzed properties show predictive control over the final performance of a model in discriminative vision tasks, generalizing across multiple network architectures.
>
> However, we agree with the reviewer that defining properties of "good normalizers" in other contexts such as NLP applications can be very useful! The primary hurdle is that for different data modalities, the standard architecture families and their corresponding optimization difficulties vary widely. For example, BatchNorm normalizes a neuron across different inputs, but input sequences in NLP applications can be of different lengths, making it incompatible with BatchNorm [1]. This issue primarily arises in RNNs-based architectures. Furthermore, in both LSTM and transformer architectures, an often noted training difficulty arises from large gradient norms, which can result in divergent training or training restarts [2, 3]. In fact, optimizers in existing NLP frameworks have gradient clipping enabled by default to avoid this problem [4]. **Arguably, to understand the effectiveness of LayerNorm in NLP, one can use our analysis in Section 5, where we showed use of LayerNorm results in suppression of the gradient norm during backpropagation!** From this perspective, one can see that LayerNorm helps reduce the training divergence produced by large gradients. Note that this indicates one can use GroupNorm with large group-size (to suppress large gradients) as well; however, since large group-size is essentially equal to LayerNorm, it is possible one may not see any added benefits. Overall, for establishing a detailed understanding of the role of normalization in NLP and other scenarios, an extensive investigation similar to our current study will be needed. For example, beyond large gradients, unbalanced gradients are also known to be a training difficulty in NLP architectures [5]. We think a thorough treatment of this topic is best left to another paper due to length constraints.
>
> *Summary:* Our identified properties help define 'good normalizers' for the broad category of discriminative vision applications. Furthermore, these properties already hint at plausible justifications to understand the success/failure of normalization layers in other contexts (e.g., to explain the success of LayerNorm in NLP). However, as a caveat, we note that since network architectures and optimization difficulties vary across different settings, generalizing our analysis to other application domains (e.g., NLP), will require substantially different experiments. We think these different analyses would be difficult to integrate into a coherent paper conforming to the NeurIPS length constraint and are best treated in future work.
>
> [1] https://arxiv.org/pdf/1607.06450.pdf
> [2] https://arxiv.org/abs/1905.11881
> [3] https://arxiv.org/pdf/2104.02057.pdf
> [4] https://discuss.huggingface.co/t/why-is-grad-norm-clipping-done-during-training-by-default/1866
> [5] https://aclanthology.org/2020.emnlp-main.463.pdf
>
> > **"What is the reason for vanishing variance of SWS in Figure 5? With the correction scaling, wouldn't one expect its activation variance to be relatively stable?"**
>
> This is a great question! Please recall the derivation for the scaling factor that helps sWS avoid vanishing variance assumes the input follows a Gaussian distribution. We suspect this is the cause of vanishing variance of sWS in Figure 5. Specifically, as mentioned in the experimental setup in the appendix, to better match the natural data distributions one will use to train neural networks in practice, our variance decay study in the paper was performed on CIFAR-100 samples. However, your question prompted us to repeat the study using unit norm Gaussian signals. In this case, we found that the vanishing variance phenomenon is reduced. Specifically, in a 20-layer CNN with sWS, the activation variance decays by a factor of 45 when we use Gaussian inputs, in contrast to decay by a factor of 1000 in the case of CIFAR-100 (from the paper). This is also on the order of decay in variance for Weight Normalization (factor of 7 decay). This indicates that when the inputs do not follow a Gaussian distribution, sWS is unable to avoid a decay in activation variance. In contrast, Weight Normalization is more robust and better prevents decay in variance. We will add these plots studying variance growth for both Gaussian inputs and CIFAR-100 to the appendix in the final version of the paper as well. We have also made them available at the following anonymous link: https://bit.ly/2U5fN6T (takes a few seconds to load due to file size).
>
> *Summary:* The correction factor designed for preventing vanishing variance in sWS assumes input data have a Gaussian distribution. This doesn't hold for many representative datasets, such as CIFAR-100, in which case sWS's correction factor is unable to prevent decay in variance. We highlight that based upon the reviewer's question, we have now designed plots for variance growth using Gaussian inputs as well and we will add them to the appendix in the final version of the paper.

---

> > ### Author Response · Authors · 2021-08-09
> > **Response to Reviewer gpva (Part 2)**
> >
> > > **"This is more of a question of curiosity that I'd like to gather the authors' thoughts on: in the case where the parameters are shared across the residual blocks (i.e., fL=fL−1=⋯=f1), how will the behavior of the normalization components change? I'm mainly thinking of the formulation of Neural ODE and other implicit models, which could also take the form $y_L = y_{L-1} + N(f(y_{L-1}))$ (written in a differential, forward Euler form), but have infinite layers. Obviously, their activations did not explode, and neither did their gradient."**
> >
> > This is indeed an interesting question! Let us assume a Neural ODE with a time horizon of 1, i.e., $\frac{d}{dt} y_t = f(y_t)$, where $t \in [0, 1]$. This ODE will retrieve the ResNet model suggested by the reviewer. Specifically, assuming the range of $t$ is discretized into $L$ steps, where the $l^{\text{th}}$ step is defined as $t_l = \frac{l}{L}$ and, consequently, we have $\Delta t = \frac{1}{L}$. Then, the forward Euler discretized version of the continuous-time ODE can be written as follows: $\Delta y_{t_l} = \Delta t \times f(y_{t_l}) \implies y_{t_{l+1}} - y_{t_{l}} = \frac{1}{L} f(y_{t_{l}})$. Substituting $t_l$ with $l$, we get $y_{l+1} = y_{l} + \frac{1}{L} f(y_{l})$. Thus, we see the discretized version of our Neural ODE is equivalent to the expression for forward propagation through a ResNet with constant weights and $L$ layers, where $l$ denotes the current layer in the ResNet model and the residual signal is scaled by a factor of $\frac{1}{L}$. Now, if we denote the variance of the $l^{\text{th}}$ layer as $var(y_{l})$, we get $var(y_{l+1}) = var(y_{l}) + \frac{1}{L^2} var(f(y_{l})) = \left(1 + \frac{1}{L^2}\right) var(y_{l})$. Here we used the standard assumptions that the residual function $f$ is initialized to be variance-preserving and the covariance between residual signal and skip path is zero. This relationship can be recursively solved to describe the output variance in terms of input variance as follows: $var(y_{L}) = \left(1 + \frac{1}{L^2}\right) var(y_{L-1}) = \left(1 + \frac{1}{L^2}\right)^{2} var(y_{L-2}) = \dots = \left(1 + \frac{1}{L^2}\right)^{L} var(y_{0})$. That is, $var(y_{L}) = \left(1 + \frac{1}{L^2}\right)^{L} var(y_{0})$. Under the assumption of infinite depth, which will retrieve the continuous-time limit for our Neural ODE, we have the following: $\lim_{L \to \infty} var(y_{L}) = \lim_{L \to \infty} \left(1 + \frac{1}{L^2}\right)^{L} var(y_{0}) = var(y_{0}) \lim_{L \to \infty} {\rm e}^{\frac{1}{L}} = var(y_{0})$. *That is, by construction, Neural ODEs are variance preserving!*
> >
> > The above derivation helps explain why Neural ODEs do not suffer from the fatal flaws of exploding variance (and consequently exploding gradients). Specifically, we recall the result by Hanin and Rolnick [1], who show the primary criterion to guarantee stable training in ResNets is to ensure stable forward propagation. Our derivation shows the ResNet network corresponding to a forward Euler discretization of the Neural ODE does not suffer from variance explosion, even under infinite depth limit. Since this infinite depth limit retrieves our continuous-time ODE itself, we reach the conclusion that a Neural ODE does not suffer from variance explosion and, consequently, can be trained effectively [1].
> >
> > *Summary:* Discretization of Neural ODEs shows the corresponding ResNet preserves the input signal's variance, hence witnessing stable training [1]. When we take infinite-depth limit of this ResNet, we retrieve our original continuous-time Neural ODE. This helps us explain why Neural ODEs do not suffer from fatal flaws of exploding or vanishing variance.
> >
> > We humbly point out to the reviewer that we are not properly familiar with the literature on Neural ODEs and skimmed a few papers to specifically answer this question. If there are any flaws in our explanation, please do let us know and we will try our best to improve the answer!
> >
> > [1] https://arxiv.org/pdf/1803.01719.pdf
> >
> >
> > > **"Where is the correlation analysis of |∇YL(J)| vs. ∏l=10L+1σN mentioned in L340-341? I didn't find it in the appendix."**
> >
> > Thank you very much for pointing this out! We sincerely forgot to add this to the appendix. We will definitely make sure to add the figure to the final version of the paper. Currently, we have made the figure available for viewing via the following anonymous link: https://bit.ly/37wT8Do (takes a few seconds to load due to file size). To summarize the plot, we note the correlation is between **0.6--0.9** in the first approximately **250 iterations** of training. After that, the correlation reduces to around 0.3 by the 350th iteration.
> >
> >
> > > **Minor points:**
> > **L302-303: Isn't the takeaway that smaller group size yield more discriminative features?**
> > **L319: from layer L to layer L−1.**
> > **L341: σN(Xl).**
> > **L594: $\mathbf{W}L \in \mathbb{R}^{D_L \times D{L-1}}$.**
> > **L606: You missed a 1N in diag in the expression of σ(XL).**
> >
> > Thank you very much for pointing out these typos! We have fixed them and they will be reflected in the final version.

---

### Official Review · Reviewer_dHUE · 2021-07-16

**Rating:** 7
**Confidence:** 4

**Summary:**

This work focuses on the comparison of different normalization techniques for deep learning, using both theoretical and empirical means.  Therefore, the contributions are in the form of novel insights, rather than a novel methodology as such.

**Limitations And Societal Impact:**

I do not see this field as being applicable in this instance.

**Main Review:**

The focus of this paper is perfectly sound and relevant, and well motivated.  The approach to analysis is convincing, the discussion and arguments are insightful, and the empirical evidence is appropriate.  The quality of writing is also good.  My only concerns as regards its acceptance has to do with novelty which I am not sure is significant enough for NIPS.  This is a subjective judgement of course and a relative call (relative to the quality of the submission I expect to be considered, or from the point of view of Chairs who have better visibility, that are considered).  I also think that the authors' observation regarding rank collapse and the activations for different input samples becoming sufficiently similar to each other that they eventually become indistinguishable in deeper layers, which is correct, should be contextualized better.  In particular, the authors should highlight the alternatives, namely GhostNorm and  SeqNorm ((https://arxiv.org/abs/2007.08554) that combines GhostNorm and GroupNorm, seems to be a promising options for avoiding rank collapse while simultaneously conferring the optimization benefits that come with BatchNorm.

Post rebuttal:

I find the authors' responses to all feedback convincing.  As regards those aimed at my initial review specifically, they do present a good case  for a greater significance of their work than I originally recognized, so I am happy to upgrade my recommendation somewhat.

**Time Spent Reviewing:**

1.5

---

> ### Author Response · Authors · 2021-08-09
> **Response to Reviewer dHUE**
>
> > **"The focus of this paper is perfectly sound and relevant, and well motivated. The approach to analysis is convincing, the discussion and arguments are insightful, and the empirical evidence is appropriate. The quality of writing is also good. My only concerns as regards its acceptance has to do with novelty which I am not sure is significant enough for NIPS. This is a subjective judgement of course and a relative call"**
>
> Thank you for your feedback! To highlight the novelty of our work, in the following we describe how our work contributes fundamental theoretical and empirical insights that help ground the heuristics often utilized for designing normalization layers in deep learning. In our opinion, this will especially be of interest to the NeurIPS community.
>
> **Contributions and Novelty of our work:** We first stress that current research on design of novel normalization techniques heavily relies on expensive empirical benchmarking to determine if their proposed techniques are better than BatchNorm and other alternatives. However, empirical benchmarking often does not generalize to different datasets or training configurations. In contrast, if it were possible to understand how underlying properties of normalization layers relate to their performance for entire application domains, it would be possible to draw general conclusions on which normalization techniques have promise and the circumstances in which they should be used. To the best of our knowledge, **our paper represents the first grounded step in this direction: we theoretically establish and empirically validate key properties that underpin the effectiveness of normalization techniques *beyond* BatchNorm.** We think our analysis will be of direct value to practitioners, who can use the principles we have identified to systematically determine which normalization technique is best suited to their specific use case, helping reduce reliance on noisy empirical benchmarking. Furthermore, we note that **our work *generalizes* known beneficial mechanisms that help justify BatchNorm's better training dynamics to several other normalization layers.** This itself is an important contribution, because prior research has focused *only* on understanding the benefits of BatchNorm, not determining which other normalization techniques provide similar benefits as BatchNorm, and in which circumstances. For example, our generalized analysis allows us to explain why GroupNorm is a highly effective normalization layer: when the group-size is properly tuned, we show GroupNorm not only helps generate discriminative features similar to BatchNorm, but also avoids the issue of gradient explosion that harms *small batch-size* training of deep CNNs with BatchNorm [1]. Further, in the context of moderate depth CNNs and ResNets, **this contribution also indicates that there are yet unknown properties underpinning the success of BatchNorm**, and an important question can be as substantial a contribution as an answer. Finally, beyond the principled unification of normalization layers, **our work also opens doors for many new and exciting discoveries**. Consider, for example, EvoNorms [2], which were designed by training *thousands of models* using an evolutionary algorithm. Using our analysis, one can directly optimize properties that result in good normalization layers without the need for training any models at all! This is in fact reminiscent of zero-cost proxies in neural architecture search [3].
>
> [1] https://openreview.net/forum?id=SyMDXnCcF7
> [2] https://arxiv.org/pdf/2004.02967.pdf
> [3] https://arxiv.org/pdf/2101.08134.pdf
>
> > **"In particular, the authors should highlight the alternatives, namely GhostNorm and SeqNorm (https://arxiv.org/abs/2007.08554) that combines GhostNorm and GroupNorm, seems to be a promising options for avoiding rank collapse while simultaneously conferring the optimization benefits that come with BatchNorm."**
>
> Thank you for this suggestion. We agree that the paper is relevant and will add discussion of it to Section 4, where the concept of rank collapse is discussed.
>
> -----------------------------------------------------------------
> **Overall Summary:** We'd like to thank you for your comments and suggestions that have helped us better contextualize our contributions. As we argue above, our work helps (i) theoretically and empirically ground often used heuristics for designing normalization layers in deep learning, (ii) generalizes existing theory specific to BatchNorm to alternative normalization layers, and (iii) opens avenues for principled design and systematic selection of novel normalization layers in deep learning. *In case our provided answers justifiably address your concerns, we respectfully request you to increase your score to support the acceptance of our work.*

---

### Official Review · Reviewer_vUUS · 2021-07-16

**Rating:** 7
**Confidence:** 4

**Summary:**

Normalization methods are effective optimization hacks that allow training deep neural networks. In this paper, different normalization methods for training neural networks are collectively studied. The focus of the paper is mostly on designing experiments that show how normalization methods allow faster training for deep neural networks.  In particular, they are studied from two main aspects:
1. The change of gradient norm (at the random initialization) when the network grows in depth.
2. The similarity of hidden representations when weights are initialized randomly.


(1) It is experimentally and theoretically shown that the normalization method can be classified into two groups: one suffering from the exponential gradient vanishing, while the other group is more robust against in terms of the norm of the gradient (in presence/absence of residual connections).


(2) First it is shown that the similarity of hidden representation (measured by pair-wise cosine similarity of inputs) is well correlated with the optimization speed. Therefore, one can predict the performance of optimization methods by this metric. Recent results confirm that BN provides distinctive features when the weights are initialized randomly. Here it is experimentally shown that this results in more normalization techniques such as group normalization.
Finally, the issue of the decay of norm of gradient is more carefully studied where it is shown that the norm of gradient although do not exponentially decay (or explodes), yet may decay through the layers which decay rate determines the optimization speed of normalization methods.

**Limitations And Societal Impact:**

Yes

**Main Review:**

The paper is well written and the results are insightful for the optimization of deep neural networks.
I have a few comments regarding the results:
1. It seems that two main factors determine the optimization performance of normalization methods: the similarity and the norm of gradient. I wondered whether the authors consider combining these two factors and reach unified predictive statistics for the optimization performance (i.e. reproducing figure 6 by also considering the norm of gradients?)
2. Is figure 6 specific to normalization methods. One can replicate the result of Fig. 6 for a deep vanilla network without any normalization layers such that we can control the cosine similarity and the norm of gradient, then check whether we can predict the performance of gradient descent from the similarity and the norm of gradient.
3. A recent paper theoretically studies the cosine similarity measure that is used in plot 6 (see "Batch Normalization Orthogonalizes Representations in Deep Random Networks").
4. I could not fully understand the reason for the gradient norm change in section 5. Since the induction over equation 1 is complicated, I can not see why this formula implies the decay/increase in the gradient norm.
5. Does these different normalizations exhibit different sensitivity to the learning rate or the size of mini-batch for stochastic optimization (not the size of the normalization batch which is studied in the paper)?

**Post Rebuttal**

I read the authors' responses. They were very detailed. Based on my understanding of the result and also other reviews and responses, I keep my score 7.

**Time Spent Reviewing:**

1

---

> ### Author Response · Authors · 2021-08-09
> **Response to Reviewer vUUS (Part 1)**
>
> > **"It seems that two main factors determine the optimization performance of normalization methods: the similarity and the norm of gradient. I wondered whether the authors consider combining these two factors and reach unified predictive statistics for the optimization performance (i.e. reproducing figure 6 by also considering the norm of gradients?)"**
>
> Thank you for this question! We agree that modeling the effects of the *similarity of activations* and *gradient explosion* in a unified manner may lead to new insights and have now conducted appropriate experiments for the same. We first describe the experimental setup, followed by empirical results and discussion.
>
> *Setup:* To analyze the two properties in a unified manner, we will need the ability to instantiate multiple models which have noticeably different degrees of gradient explosion and similarity of activations. This can be achieved by varying the group-size in GroupNorm. Specifically, as shown in the paper, increasing the group-size *increases* the similarity of activations in deeper layers, while *reducing* the degree of gradient explosion in earlier layers. We use 20-layer CNNs trained on CIFAR-100 for our experiments. The degree of gradient explosion will be measured by fitting an exponential curve ($e^{ax}$) to the gradient norms at different convolutional layers ($x$, in $e^{ax}$). A large, positive value of the exponent $a$ indicates more gradient explosion in earlier layers. For similarity of activations, we measure cosine similarity of activations produced by different inputs at the final convolutional layer. Optimization speed is determined by measuring mean training accuracy after a consistent number of training epochs, as done in the paper. Results are shown in the table below. All models are trained with a batch size of 256 at learning rate of 0.1 for 60 epochs. Results are averaged over 3 seeds.
>
> |     Group-Size     |   1   |   2   |   4   |   8   |   16   |   32   |   64   |
> | ------------------ |-------|-------|-------|-------|--------|--------|--------|
> |   Mean Train Acc.  | 47.8% | **59.8%** | 55.2% | 43.9% | 37.5%  |  36.7% |  35.7% |
> |   Cos. Similarity  |  0.61 | **0.71**  |  0.83 |  0.91 |  0.96  |  0.96  |  0.98  |
> |   Grad. Exp. ($a$) |  1.65 | **-0.05**  | -0.86 | -1.67 | -1.97  |  -2.32 |  -2.48 |
>
> We have also made available a figure corresponding to the above table at the following anonymous link: https://bit.ly/3lJJyp1 (takes a few seconds to load due to file size).
>
> *Discussion:* The results in the table above show that the optimization speed first improves with increasing group-size, achieves a maximum, and then starts decreasing with further increase in group-size. *This behavior is in fact implied by the analysis in our paper!* Specifically, Section 5 shows that increasing the group-size will reduce the severity of gradient explosion, hence improving training dynamics and consequently optimization speed. However, Section 4 shows that when the group-size is increased, similarity of activations also increases, leading to reduced optimization speed (see Figure 6). Combining these two results, our analysis predicts that the group-size in GroupNorm controls a speed-stability trade-off: with increase in group-size, the optimization speed should first improve (due to improved training stability), achieve a maximum, and then decrease (due to increased similarity of activations). This predicted behavior is consistent with the empirical results reported in the table above!
>
> *Summary:* Changing group-size in GroupNorm enables the similarity of activations and gradient explosion to be modeled in a unified manner, exposing a optimization speed vs. training stability trade-off. We have converted the above table into a figure and will add it to the final version of the paper as well.
>
>
> > **"Is figure 6 specific to normalization methods. One can replicate the result of Fig. 6 for a deep vanilla network without any normalization layers such that we can control the cosine similarity and the norm of gradient, then check whether we can predict the performance of gradient descent from the similarity and the norm of gradient."**
>
> *To the best of our knowledge, a predictive characterization of optimization speed using similarity of activations has not been done in prior deep learning research*. However, we agree with the reviewer's expectation and believe that our finding in figure 6 can be generalized outside of normalization layers as well. To justify this, we note that the Neural Tangent Kernel [1], which can predict optimization and generalization characteristics of wide neural networks, has recently been shown to be highly related to a kernel defined by the inner products of activations, i.e., the similarity of activations [2]. This indicates using properties that allow one to tune similarity of output activations should allow control over behavior of optimization speed (given other factors remain the same), resulting in similar plots as Figure 6.
>
> *Summary:* We expect any property that helps control similarity of activations should allow one to create plots like Figure 6. This expectation can be partially justified by relating the Neural Tangent Kernel [1] with a kernel defined by the similarity of activations [2]. We will cite this latter work in our final version, but please note that it was released *after the NeurIPS deadline* and therefore could not have been cited in the submitted version.
>
> [1] https://arxiv.org/abs/1806.07572
> [2] https://arxiv.org/pdf/2106.08453.pdf
>
> > **"A recent paper theoretically studies the cosine similarity measure that is used in plot 6 (see "Batch Normalization Orthogonalizes Representations in Deep Random Networks")."**
>
> Thank you for suggesting this reference! We will cite it. Note that it was publicly released *after the NeurIPS deadline* and therefore could not have been cited in the submitted version.
>
> > **"I could not fully understand the reason for the gradient norm change in section 5. Since the induction over equation 1 is complicated, I can not see why this formula implies the decay/increase in the gradient norm."**
>
> Thank you for this comment! In the following, we first provide a brief overview of our analysis of Equation 1, via which we identified the source of gradient explosion in normalization layers in a general manner. This is followed by an illustrative example.
>
> Equation 1 gives the expression for how the gradient is modified as it propagates backward through the $L^{th}$ normalization layer. Specifically, the expression shows that in contrast with a vanilla neural network, during backprop in a BatchNorm model, the gradient of any given layer is divided by the standard deviation of the channel's pre-activations. At initialization, assuming Gaussian inputs, a channel's pre-activations have an expected standard deviation of $\sqrt{\frac{\pi-1}{\pi}} = 0.82$ (i.e., less than 1). *This implies, due to division by standard deviation during backward propagation, the gradient is amplified according to the channel statistics calculated during forward propagation*. Since all normalization layers will perform this gradient amplification, extending this analysis to multiple layers, one can see the root cause of gradient explosion is division by standard deviation (which is less than 1 at initialization).
>
> Consider the following illustrative example for understanding this more intuitively. Denote gradient at layer $l$ as $g_{l}$ and the inverse of the standard deviation of pre-activations as $K = \frac{1}{\sigma_{l}}$. Then, we have $\frac{||g_{l-1}||}{||g_{l}||} \propto K$. For the sake of this example, assume the standard deviation of pre-activations at any given layer is constant (under proper initialization, this will be true for gaussian inputs and constant-width models). Then, in an $L$-layer model, we have $\frac{||g_{1}||}{||g_{L}||} \propto K^{L}$. Since the expected standard deviation at any given layer is less than 1, we have $K > 1$. *This implies the ratio of the gradient norm at the first layer to the last layer increases exponentially with model depth, resulting in gradient explosion in earlier layers!*.
>
> *Summary:* Division of the gradient by standard deviation of activations, which is less than 1 at initialization, amplifies the gradient during backward propagation from a normalization layer. This amplification is exponential in the number of layers, leading to gradient explosion in deep networks.

---

> > ### Author Response · Authors · 2021-08-09
> > **Response to Reviewer vUUS (Part 2)**
> >
> > > **"Do these different normalizations exhibit different sensitivity to the learning rate or the size of mini-batch for stochastic optimization (not the size of the normalization batch which is studied in the paper)?"**
> >
> > In our experiments, we have seen that almost all normalization layers except for BatchNorm and EvoNormB0 are quite sensitive to large learning rates (e.g., see Figure 1 and appendix, where we show train/test curves for different learning rates). In general, we believe better understanding the robustness of BatchNorm to large learning rates, which are known to have beneficial regularization effects [1], can further help ascertain how a novel normalization layer can inherit the regularization properties of BatchNorm. Specifically, note that several normalization layers will induce scale-invariance of filter weights, which provably enables convergence via an auto-tuning of the learning rates if the initial learning rate is less than a certain constant defined by a combination of Lipschitz constants of scale-variant and scale-invariant parameters [2]. That is, as long as the initial learning rate is smaller than an upper bound, training provably converges in scale-invariant models. This indicates the sensitivity to initial learning rate magnitude can tell a lot about the effectiveness of normalization layers. In fact, Karakida et al. [3] provably illustrate BatchNorm's robustness to large learning rates by analyzing the Fisher Information Matrix. For alternative normalization layers, we expect their loss landscape is not properly conditioned, resulting in sensitivity to large learning rates. We have conducted preliminary experiments in this vein, calculating the eigenspectrum of the model Hessian, and have seen that normalization layers which are sensitive to large learning rates have either ill-conditioned or overly smooth loss landscapes (it is possible the latter may not be causally related with learning rate sensitivity). For now, we leave this direction for future work, but comment here that it may hold promise.
> >
> > *Summary:* Other than BatchNorm and EvoNorms, we found that all normalization layers are sensitive to large learning rates. A thorough analysis of this will require establishing connections with the model Hessian (or the Fisher Information Matrix). Preliminary empirical evidence indicates normalization layers sensitive to large learning rates can have either ill-conditioned or overly smooth loss landscapes. However, we believe a thorough treatment is beyond the scope of the current paper and leave it to future work.
> >
> > [1] https://arxiv.org/abs/2003.02218
> > [2] https://openreview.net/forum?id=rkxQ-nA9FX
> > [3] https://arxiv.org/abs/1906.02926

---

> > > ### Comment · Reviewer_vUUS · 2021-08-24
> > > **Ambiguity about "Group Size"**
> > >
> > > I thank the authors for their detailed response! In our internal discussions, we noticed that the term Group size, which is frequently used in Section 4, is not clearly defined. Is **Gropu Size** is the **number of groups** denoted by $G$?

---

> > > > ### Author Response · Authors · 2021-08-24
> > > > **Group size = number of channels grouped in GroupNorm**
> > > >
> > > > Thank you for very much your question! It seems we have made a typo in lines 276 and 290, leading to the confusion. Indeed, the *group-size is equal to the number of channels grouped together for normalization in GroupNorm*. The number of groups then is layer-width divided by the group-size. This implies LayerNorm has group-size equal to layer-width and only 1 group; meanwhile Instance Normalization has group-size equal to 1 and number of groups equal to layer-width.
> > > >
> > > > In line 276, we made a typo and wrote "G is number of groups", when we intended to write "G is group-size". Similar error exists in line 290. We have fixed both the errors and now provide a clear definition for both group-size and number of groups in the text. These changes will be reflected in the final version of the paper. Again, thank you for pointing this out!

---

> > > > > ### Comment · Area_Chair_iGdt · 2021-08-24
> > > > > **A few more things**
> > > > >
> > > > > Thanks for the clarification. Just to make sure we are all on the same page, can you please further explain:
> > > > > 1) In figure 7b, the meaning of G is not "Group size" but it is the "number of groups", since it gets smaller as we get closer to layer norm, right?
> > > > > 2) Elsewhere, in lines 355-356, G also has an additional meaning, as the set of indices within the group, right?

---

> > > > > > ### Author Response · Authors · 2021-08-24
> > > > > > **Figure 7b and lines 355--356**
> > > > > >
> > > > > > Thank you for the question! Responding to the specific pointers:
> > > > > > 1. In figure 7b, it should indeed be number of groups, as you remarked. We originally wrote the paper with G defining number of groups, but realized close to the deadline that group-size is a more intuitive quantity. This has caused a few typos and we will make sure to thoroughly check for these again. As of now, we are certain that apart from the places pointed to by the AC and reviewers, there are no more location where this typo exists. We have fixed these errors in both the text and figures.
> > > > > >
> > > > > > 2. In lines 355-356, we abused the notation slightly and wrote $i \in G$, which represented filters belonging to a given group of group-size G. We see now that this was not ideal. To remedy this, we have now defined a new notation {$g$} to denote a set of $G$ filters that are grouped together for normalization. With this change, the text becomes $i \in $ {$g$}, hence avoiding overloading of the same symbols. This will be reflected in the final version of the paper.
> > > > > >
> > > > > > Again, thank you very much for pointing out these typos and helping us improve the paper!

---

> > > > > > > ### Comment · Area_Chair_iGdt · 2021-08-25
> > > > > > > **Thanks**
> > > > > > >
> > > > > > > Thanks for the clarifications. All these typos made section 4 very confusing to me. BTW, G denoted "number of groups" in the original group normalization paper, so I would not change its meaning in this paper, to avoid reader confusion.

---

> > > > > > > > ### Author Response · Authors · 2021-08-25
> > > > > > > > **Justified suggestion**
> > > > > > > >
> > > > > > > > Following GroupNorm and using G for number of groups is definitely a justified suggestion and we will make sure to follow it. We have also fixed typos in Section 4 (both figure 7 and text) to improve clarity of the section. Thank you very much for pointing those out! These changes will be reflected in the final version of the paper.

---

### Decision · Program_Chairs · 2021-09-27

**Decision:**

Accept (Poster)

**Comment:**

This paper performs extensive empirical scans, comparing various aspects of normalization methods. The reviews are all positive, but the discussion raised a focused on novelty and clarity issues. Regarding novelty, some results seem to be less surprising and seem like a validation of existing understanding (sections 3 and 5), while some results seem to be more interesting (section 4). Regarding clarity, the paper had a lot of confusing typos (especially in the more interesting section 4), but those were cleared in the discussion. Still, I feel some parts are missing details, and I recommend to author to improve it (e.g., I couldn't follow the "proof idea" of conjecture 1, or the parts with the harmonic mean in section 5).